# Pharmaceutical targeting of OTUB2 sensitizes tumors to cytotoxic T cells via degradation of PD-L1

Wenfeng Ren [1,2,6], Zilong Xu [1,2,6], Yating Chang [1,2,3,6], Fei Ju[1,2,6], Hongning Wu[1,2,3,6], Zhiqi Liang[1,2,3,6], Min Zhao[1,2], Naizhen Wang[1,2], Yanhua Lin[1,2], Chenhang Xu[1,2,3], Shengming Chen[1,2], Yipeng Rao[1,2], Chaolong Lin[1,2], Jianxin Yang[4,5], Pingguo Liu [4,5] ✉, Jun Zhang [1,2] ✉, Chenghao Huang [1,2] ✉ & Ningshao Xia [1,2,3] ✉

PD-1 is a co-inhibitory receptor expressed by CD8[+] T cells which limits their cytotoxicity. PD-L1 expression on cancer cells contributes to immune evasion by cancers, thus, understanding the mechanisms that regulate PD-L1 protein levels in cancers is important. Here we identify tumor-cell-expressed otubain-2 (OTUB2) as a negative regulator of antitumor immunity, acting through the PD-1/PD-L1 axis in various human cancers. Mechanistically, OTUB2 directly interacts with PD-L1 to disrupt the ubiquitination and degradation of PD-L1 in the endoplasmic reticulum. Genetic deletion of OTUB2 markedly decreases the expression of PD-L1 proteins on the tumor cell surface, resulting in increased tumor cell sensitivity to CD8[+] T-cell-mediated cytotoxicity. To underscore relevance in human patients, we observe a significant correlation between OTUB2 expression and PD-L1 abundance in human non-small cell lung cancer. An inhibitor of OTUB2, interfering with its deubiquitinase activity without disrupting the OTUB2-PD-L1 interaction, successfully reduces PD-L1 expression in tumor cells and suppressed tumor growth. Together, these results reveal the roles of OTUB2 in PD-L1 regulation and tumor evasion and lays down the proof of principle for OTUB2 targeting as therapeutic strategy for cancer treatment.

The ability to boost antitumor immunity within the tumor microenvironment (TME) is critical for tumor control[1]. Cancers frequently develop disparate mechanisms that allow them to evade destruction by the immune system[2]. One of the escape mechanisms involves the interaction between antitumor cytotoxic T lymphocytes (CTLs) and tumor cells. Tumor cells either resist or impair CTL attack through multiple tumor cell-intrinsic mechanisms, including upregulation of inhibitory immune checkpoint receptors[3–5], impaired tumor antigen presentation[6,7], secretion of inhibitory cytokines or ligands[8–10] and alterations in metabolic pathways[11,12], as well as other mechanisms[13,14].

[1]State Key Laboratory of Vaccines for Infectious Diseases, Xiang An Biomedicine Laboratory, Department of Laboratory Medicine, School of Public Health, Xiamen University, Xiamen, Fujian 361102, China. [2]State Key Laboratory of Molecular Vaccinology and Molecular Diagnostics, National Institute of Diagnostics and Vaccine Development in Infectious Diseases, Xiamen University, Xiamen, Fujian 361102, China. [3]School of Life Sciences, Xiamen University, Xiamen, Fujian 361102, China. [4]Department of Hepatobiliary & Pancreatic Surgery, The National Key Clinical Specialty, Zhongshan Hospital of Xiamen University, School of Medicine, Xiamen University, Xiamen, Fujian 361004, China. [5]Fujian Provincial Key Laboratory and Chronic Liver Disease and Hepatocellular Carcinoma, Xiamen, Fujian 361004, China. [6]These authors contributed equally: Wenfeng Ren, Zilong Xu, Yating Chang, Fei Ju, Hongning Wu, Zhiqi Liang. ✉e-mail: pgliu@xmu.edu.cn; zhangj@xmu.edu.cn; huangchenghao@xmu.edu.cn; nsxia@xmu.edu.cn

Binding of programmed death-1 ligand 1 (PD-L1) on tumor cells to its receptor programmed death-1 (PD-1) on T cells leads to suppression of the tumor-killing activity of T cells, representing a pivotal mechanism of immune evasion by tumor cells. Immune checkpoint blockade (ICB) targeting PD-1/PD-L1 signaling has produced remarkable clinical benefits in cancer patients with Hodgkin's lymphoma, Merkel cell carcinoma, skin cancer, non-small-cell lung carcinoma (NSCLC), or bladder cancer, as well as other cancer types[15], demonstrating the importance of regulating PD-1/PD-L1 signaling in cancer treatment.

Tumor cells exploit the expression of PD-L1 to avoid T-cell-mediated tumor immunosurveillance. In addition to its well-established role in immune regulation, PD-L1 has been reported to have intrinsic pro-oncogenic functions that may enhance the proliferation and survival of tumor cells[16]. Recent studies have revealed evidences that tumor cells may adaptively upregulate their expression of PD-L1 after treatment with anti-PD-1 antibodies, representing a mechanism of acquired resistance to ICB immunotherapy[17,18]. These findings underscore the need to dissect the mechanism underlying the regulation of PD-L1 protein levels in cancers to either improve therapeutic outcomes or enhance the ICB response. PD-L1 expression can be induced in response to inflammatory cytokines, such as interferon-γ (IFN-γ), secreted by immune cells within the TME or can be driven by tumor cell-intrinsic mechanisms, including transcriptional activation of PD-L1[19,20], increased efficiency of intracellular PD-L1 trafficking via escape from lysosome-mediated degradation[21,22] and stabilization of PD-L1 by posttranslational modifications[23–26]. Although it is evident that the ubiquitination-dependent degradation pathway may control the fate of the PD-L1 protein, the exact mechanisms underlying the endoplasmic reticulum (ER)-associated degradation (ERAD) of PD-L1 remain unclear. These findings motivated us to identify new regulators of both constitutive and induced cell-surface PD-L1 expression to reprogram the TME for cancer treatment.

Otubain-2 (OTUB2) is a deubiquitinating enzyme (DUB) that belongs to the ovarian tumor (OTU) superfamily of proteins[27]. OTUB2 is expressed at high levels in human malignancies and at variable lower levels in organs throughout the body (https://www.proteinatlas.org/ENSG00000089723-OTUB2/tissue). Its genetic activation can induce cancer metastasis in certain cell types[28,29], promotes liver cancer cell growth[30], inhibit the cellular antiviral response[31], fine-tune the choice of DNA repair pathways[32], and influence osteogenic differentiation[33]. Although extensive investigations into the molecular biology of this protein have been performed, there has been no substantive documentation on the potential of OTUB2 to play an immunoregulatory role in T cells. To our knowledge, no specific clinically relevant OTUB2 inhibitor is available.

In the present study, we show that OTUB2 is a key regulator of PD-L1 in a broad range of tumor cells. OTUB2 interacts with PD-L1 in the ER and to disrupt the ERAD of PD-L1 to impair T-lymphocyte-mediated antitumor immunity. On the structural basis of the binding pocket of OTUB2, we identify OTUB2-IN-1, a specific inhibitor of OTUB2, which can reduce the expression of PD-L1 proteins in tumor cells and suppress tumor growth by promoting robust intra-tumoral infiltration of CTLs. Taken together, our data suggest that OTUB2 may be a potential therapeutic target for cancer immunotherapy.

## Results

### OTUB2 levels negatively correlate with antitumor CTL signatures and prognosis in multiple cancers

To explore DUBs that may regulate the antitumor immune responses of CD8+ T cells in the TME, we performed correlation analyses specifically for the CD8+ T-cell signature gene *CD8A* with 103 DUB genes[34] using The Cancer Genome Atlas (TCGA) data sets (Supplementary Table 1). We found *OTUB2* to be the top negative regulator gene among all analyzed DUB genes, which showed negative associations with the CD8+ T-cell signature gene *CD8A* in lung squamous carcinoma (LUSC)

and stomach adenocarcinoma (STAD) (Supplementary Fig. 1a). Notably, OTUB2 were significantly upregulated in the cancer tissues of these two cancer types compared with adjacent paracancerous tissues (Supplementary Fig. 1b), suggesting that OTUB2 may play an immunoregulatory role in the TME. To further substantiate the previously reported upregulation of OTUB2 in multiple cancers[28,29], we assessed OTUB2 expression in human cancer tissues compared with adjacent paracancerous tissues using the Tumor Immune Estimation Resource (TIMER) algorithm[35]. OTUB2 was variably upregulated in most types of human solid tumors (Supplementary Fig. 2a). Analysis of bulk tumor gene expression data from TCGA revealed a significant negative correlation between the genes encoding OTUB2 and intratumoral T-cell abundance (*CD3E* and *CD8A*) in multiple cancer types (Fig. 1a and Supplementary Fig. 2b). Similarly, we observed that across multiple cancer types, OTUB2 expression was negatively correlated with the expression of the CTL signature genes *GZMB*, *GZMA* and *CD69* (Fig. 1b and Supplementary Fig. 2c, d). Furthermore, OTUB2 expression was negatively correlated with patient overall survival (OS) and disease-free survival (DFS) in multiple cancer types (Supplementary Fig. 3a–h). Collectively, these analyses indicate that the intratumoral CD8+ CTL abundance is increased in cancer tissues with a low OTUB2 level, suggesting that OTUB2 may be involved in the regulation of the tumor-specific immune response.

### Assessing OTUB2 functions by its genetic deletion and overexpression in cancer cells

Because of their highly conserved catalytic OTU domain, OTUB2 homologs from human and mouse may display conserved function that can specifically remove ubiquitin (Ub) chains from target proteins[36]. To assess the immunoregulatory effect of OTUB2 on tumor growth in syngeneic mouse models, we first knocked down OTUB2 expression in mouse MC38 cells using short hairpin RNA (shRNA) (Supplementary Fig. 4a). Cell viability was similar between control (Scramble) and OTUB2-knockdown (KD) (sh-OTUB2) MC38 cells (Supplementary Fig. 4b). Knocking down OTUB2 reduced MC38 tumor growth and overall survival in immunocompetent C57BL/6J mice (Fig. 1c and Supplementary Fig. 4c). Furthermore, depleting CD8+ T cells with anti-CD8 antibodies reversed this OTUB2-KD-induced MC38 tumor growth inhibition in C57BL/6J mice (Fig. 1d and Supplementary Fig. 5), which functionally implicates CD8+ T cells in the OTUB2-KD tumor phenotype. To extend functional studies to other cancer types, we used CRISPR/Cas9 technology to genetically delete the *OTUB2* gene in three mouse tumor cell lines (MC38, LL/2 and B16-F10) (Supplementary Fig. 6a–c). Knocking out OTUB2 did not alter the morphology or in vitro growth rate of these tumor cells (Supplementary Fig. 6d–f). When wild-type (WT, Control) and OTUB2-knockout (KO) MC38 cells were inoculated parallel into syngeneic and immunodeficient mice, knocking out OTUB2 reduced MC38 tumor growth in immunocompetent C57BL/6J mice but not in immunodeficient BALB/c-nude mice (Fig. 1e, f). Analogous phenotypic differences in tumor growth were also observed between WT and OTUB2-KO mouse LL/2 and B16-F10 cells (Fig. 1g, h). In contrast, overexpression (OE) of OTUB2 in B16-F10 cells did not alter the in vitro growth rate of tumor cells but promoted tumor growth in C57BL/6J mice (Fig. 1i and Supplementary Fig. 6g, h). Collectively, these data demonstrate that OTUB2-KO tumors grow more slowly in immunocompetent mice but not in immunodeficient mice, which suggests the involvement of the adaptive immune system.

### OTUB2 interacts with the intracellular domain (ICD) of PD-L1

We next investigated the molecular mechanism by which OTUB2 is involved in the regulation of the tumor-specific immune response. Analysis of the OTUB2 interactome by mass spectrometry revealed PD-L1 as the binding partner of OTUB2 in B16-F10 cells (Supplementary Fig. 7a). In line with this result, ectopically expressed OTUB2

associated with PD-L1 in HEK293T cells (Fig. 2a), and E. coli-purified OTUB2 interacted with PD-L1 in vitro (Fig. 2b and Supplementary Fig. 7b). Ectopically expressed OTUB2-WT but not catalytically inactive OTUB2 (C51S mutant) associated with PD-L1 in HEK293T cells (Fig. 2c). We also observed that endogenous OTUB2 interacted with PD-L1 in multiple cell lines by using primary antibodies against either OTUB2 or PD-L1 (Fig. 2d and Supplementary Fig. 7c–e). To determine the subcellular location of the OTUB2-PD-L1 interacting complexes, we performed a Duolink in situ proximity ligation assay (PLA) and observed that the PLA signals were localized mainly in the cytoplasm (Fig. 2e and Supplementary Fig. 7f, g). Next, we characterized the interaction between OTUB2 and PD-L1. A coimmunoprecipitation (co-IP) assay using mutant PD-L1 or OTUB2 proteins with different structural

domains deleted revealed the involvement of the ICD (261–290) of PD-L1 in the interaction with OTUB2 and the OTU domain (44–234) of OTUB2 in the interaction with PD-L1 (Fig. 2f, g).

## Identification of OTUB2 as a positive regulator of PD-L1

Because of the well-known role of OTUB2 acting as a deubiquitinase in regulating the levels of proteins, we hypothesized that OTUB2 may regulate the expression of PD-L1. We first examined the effect of OTUB2 deficiency on PD-L1 levels on the surface of lung cancer cells. Flow-cytometry analyses indicated that the expression of PD-L1 on the surface of tumor cells was significantly reduced in OTUB2-KD or OTUB2-KO cells compared with control cells (Fig. 3a–c and Supplementary Fig. 8a, b). Because IFN-γ is regarded as a major inducer of PD-

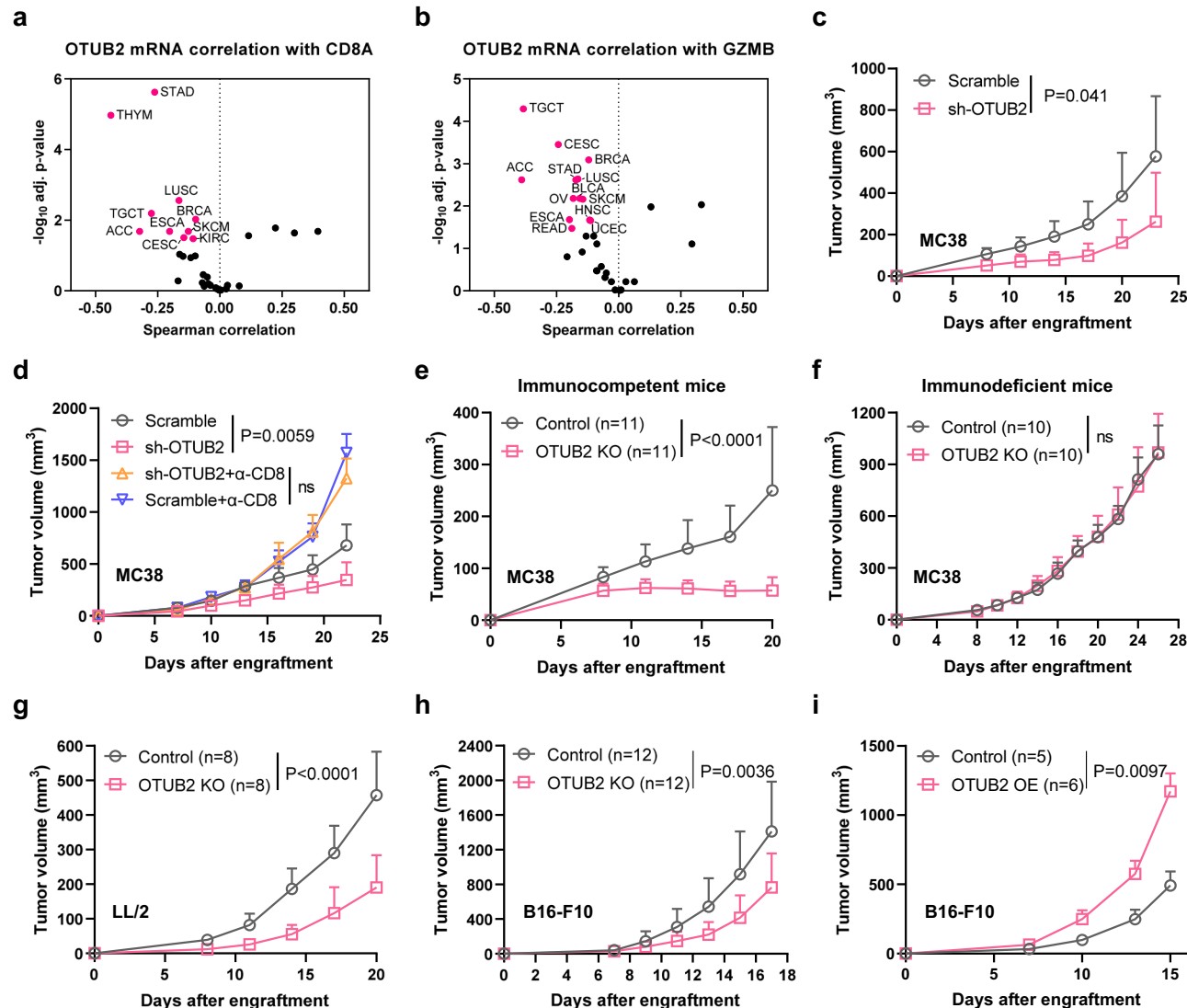

**Fig. 1 | Identification of OTUB2 as a potent promoter of cancer immune evasion.** Associations between OTUB2 expression and transcriptomic signature of CD8⁺ T-cell-associated genes. **a** Volcano plot of the Spearman correlation between *OTUB2* and *CD8A* mRNA expression in 32 cancer types. Magenta points indicate cancer types in which *OTUB2* was significantly negatively correlated with *CD8A* (adjusted $P < 0.05$). **b** Volcano plot of the Spearman correlation between *OTUB2* and *GZMB* mRNA expression in 32 cancer types. Magenta points indicate cancer types in which *OTUB2* was significantly negatively correlated with *GZMB* (adjusted $P < 0.05$). **c** Tumor volume over time in C57BL/6 mice implanted with vector control or OTUB2-KD MC38 tumor cells. $n = 5$ mice per group. **d** Tumor growth of vector control or OTUB2-KD MC38 tumor cells in C57BL/6 mice depleted of CD8⁺ T cells. $n = 6, 6, 9$ and 10 mice in the four groups. **e** Tumor volume over time in C57BL/6

mice implanted with vector control or OTUB2-KO MC38 tumor cells. $n = 11$ mice per group. **f** Tumor volume over time in immunodeficient mice implanted with vector control or OTUB2-KO MC38 tumor cells. $n = 10$ mice per group. **g** Tumor volume over time in C57BL/6 mice implanted with vector control or OTUB2-KO LL/2 tumor cells. $n = 8$ mice per group. **h** Tumor volume over time in C57BL/6 mice implanted with vector control or OTUB2-KO B16-F10 tumor cells. $n = 12$ mice per group. **i** Tumor volume over time in C57BL/6 mice implanted with vector control or OTUB2-OE B16-F10 tumor cells. $n = 5$ and 6 mice in the two groups. The data are presented as the mean ± s.e.m. from at least two independent experiments, and $P$ values were calculated using two-way analysis of variance (ANOVA) (**c–i**), except for in (**a, b**), in which $P$ values were calculated using Spearman's correlation test. ns, not significant.

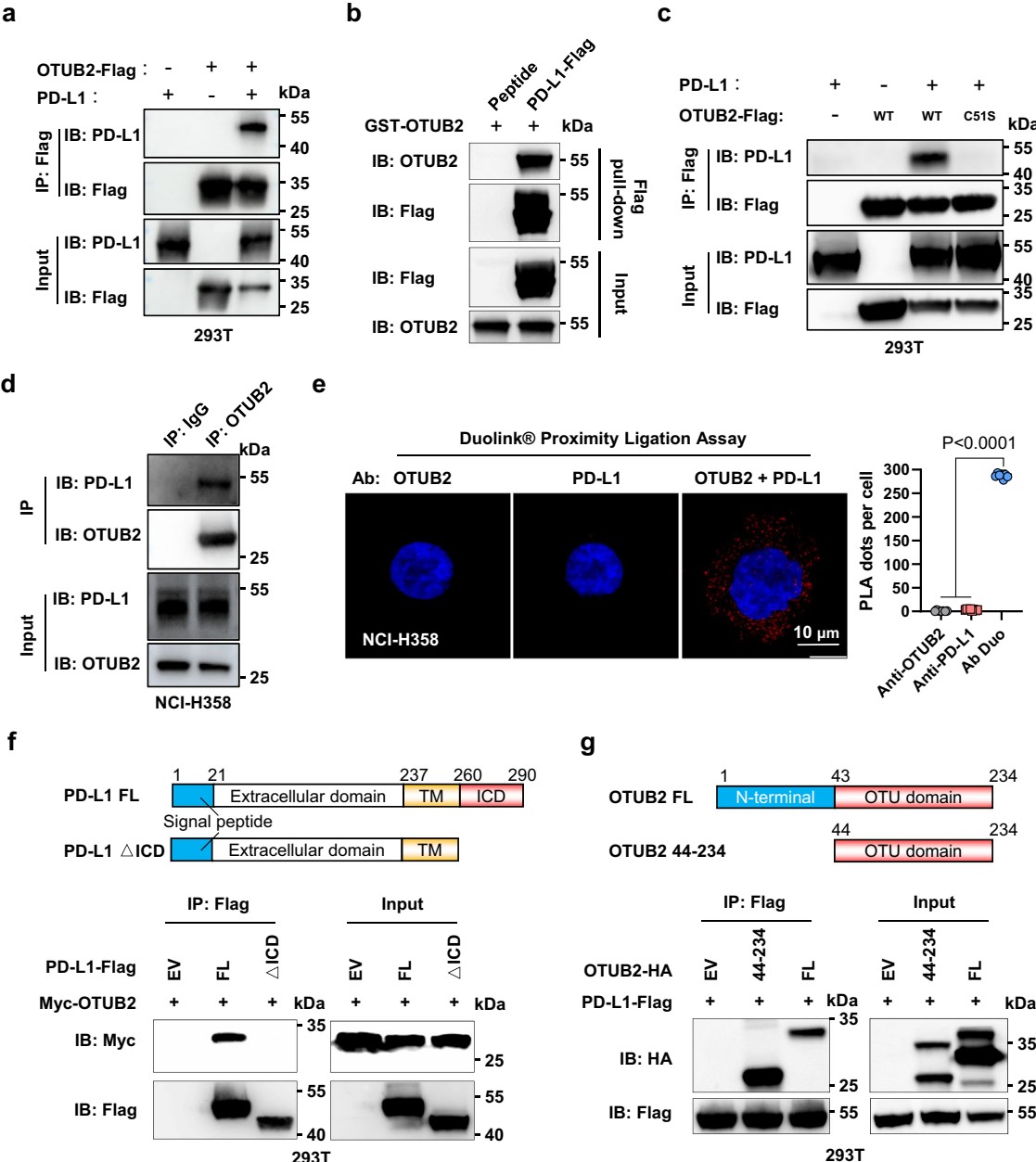

**Fig. 2 | OTUB2 specifically interacts with the intracellular domain (ICD) of PD-L1 in vivo and in vitro. a** 293 T cells were cotransfected with OTUB2-Flag and untagged PD-L1 constructs, and immunoprecipitation was performed with anti-Flag antibody to examine the interaction between OTUB2-Flag and PD-L1. **b** The in vitro interaction between purified PD-L1-Flag proteins and GST-OTUB2 proteins examined by in vitro Flag pull-down assays. **c** 293 T cells were cotransfected with wild-type (WT) OTUB2-Flag or the C51S mutant along with a PD-L1 construct, and immunoprecipitation was performed with an anti-Flag antibody to examine the interaction between the OTUB2 mutant and PD-L1. **d** Immunoprecipitation was performed with an anti-OTUB2 antibody to examine the interaction between endogenous OTUB2 and PD-L1 in NCI-H358 cells. **e** In situ interaction between OTUB2 and PD-L1 in NCI-H358 cells. After fixation, a Duolink in situ PLA was

performed with anti-OTUB2 and anti-PD-L1 antibodies to assess the OTUB2-PD-L1 interaction. Quantification of the PLA dots indicating PD-L1-OTUB2 interactions is shown as the mean ± s.e.m. Scale bar, 10 μm. $n = 8$ biologically independent samples. **f** Schematic representation of full-length (FL) and ΔICD PD-L1 constructs. The interactions between Myc-OTUB2 and PD-L1 FL or ΔICD were detected by a co-IP assay. **g** Schematic representation of OTUB2 FL and truncated (aa 44-234) constructs. The interactions between PD-L1-Flag and OTUB2 FL or aa 44-234 were detected by a co-IP assay. The experiments were repeated at least twice independently with similar results. The data are presented as the mean ± s.e.m., and $P$ values were calculated using an unpaired two-sided Student's $t$ test (**e**). Source data are provided as a Source Data file.

L1 upregulation within the TME, we then tested the effect of OTUB2 on PD-L1 expression in the presence of IFN-γ stimulation. Notably, knocking out or knocking down OTUB2 significantly attenuated the induction of PD-L1 expression by IFN-γ (Fig. 3a–c). The effects of OTUB2 deficiency on PD-L1 levels were also found in other human or mouse tumor cell lines such as MDA-MB-231 breast cancer, LoVo colon cancer and B16-F10 melanoma cancer cells (Supplementary Fig. 8c–h).

In contrast, OTUB2 OE in lung cancer cells enhanced both endogenous PD-L1 expression and IFN-γ-induced PD-L1 expression (Fig. 3d–f and Supplementary Fig. 9a, b). Consistently, introduction of OTUB2-WT but not OTUB2-C51S enhanced both endogenous PD-L1 expression and IFN-γ-induced PD-L1 expression (Fig. 3d–f and Supplementary Fig. 9a, b). The effects of OTUB2 overexpression on PD-L1 levels were also found in other human or mouse tumor cell lines such as LoVo

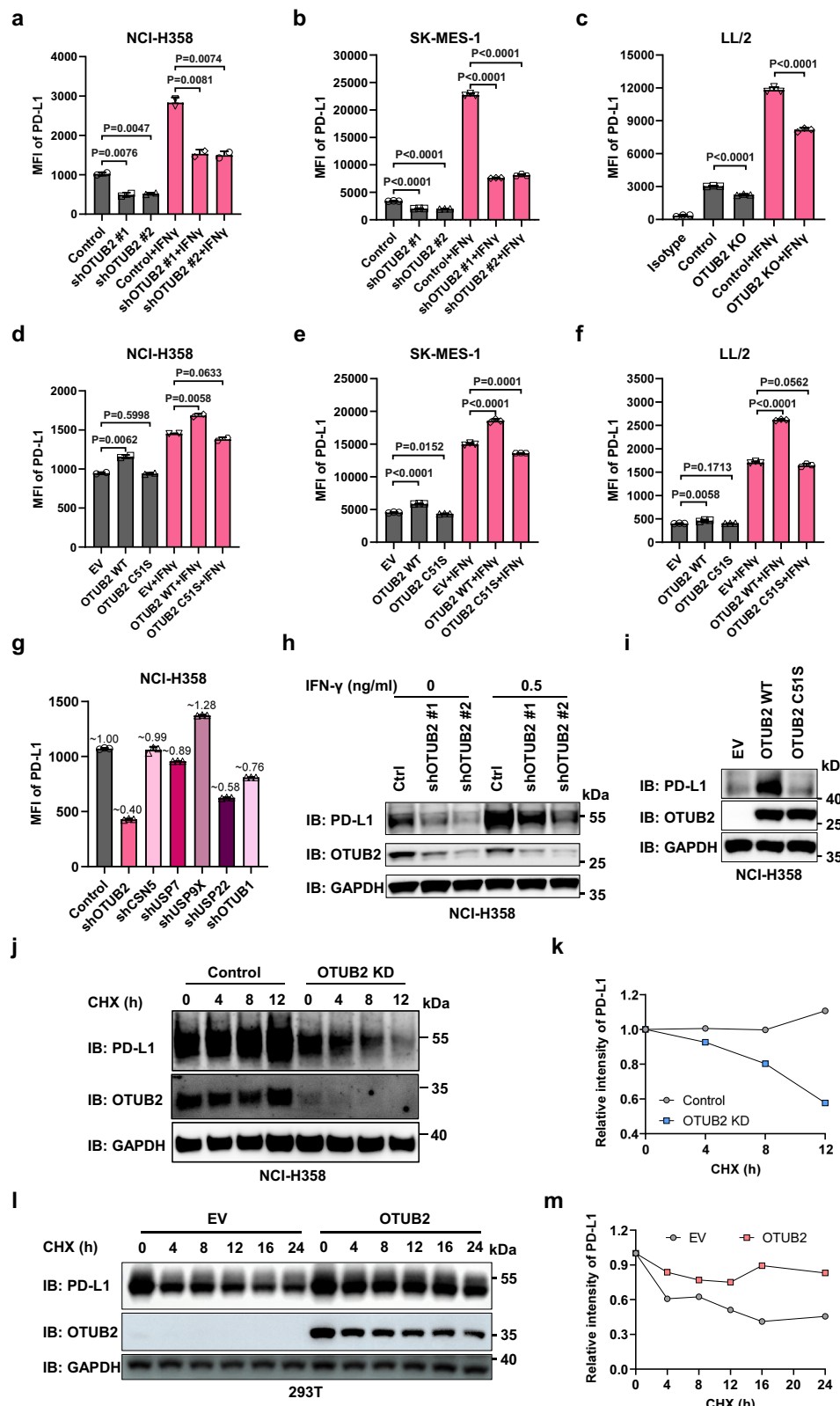

colon cancer and B16-F10 melanoma cancer cells (Supplementary Fig. 9c–e). At least five DUBs were reported to stabilize PD-L1 in different cancers, including USP7, USP9X, USP22, CSN5 and OTUB1[23,37–40]. Although USP7, USP22 and OTUB1 played a certain role in the regulation of PD-L1 in NCI-H358 cells and USP22 slightly regulated PD-L1 abundance in SK-MES-1 cells, we found that OTUB2 had the most significant impact on the regulation of PD-L1 both in NCI-H358 and SK-

MES-1 lung cancer cells (Fig. 3g and Supplementary Fig. 10a–g). We further examined the effect of OTUB2 deficiency on the total PD-L1 level in tumor cells. Western blot analysis of cellular lysates confirmed that OTUB2 KD or KO induced a decrease in the PD-L1 protein level, while OTUB2-WT but not OTUB2-C51S overexpression caused an increase (Fig. 3h, i and Supplementary Fig. 11a–f). The cycloheximide (CHX)-chase experiment demonstrated faster degradation of PD-L1

**Fig. 3 | OTUB2 stabilizes PD-L1 in tumor cells. a, b** Flow cytometry analysis of PD-L1 levels on the surface of control and OTUB2-KD NCI-H358 or SK-MES-1 cells treated with or without IFN-γ. MFI, median fluorescence intensity. *n* = 2 biologically independent experiments (**a**), *n* = 3 biologically independent experiments (**b**). **c** Flow cytometry analysis of PD-L1 levels on the surface of control and OTUB2-KO LL/2 cells treated with or without IFN-γ. *n* = 3 biologically independent samples. **d, e** Flow cytometry analysis of PD-L1 levels on the surface of control and OTUB2-OE NCI-H358 or SK-MES-1 cells treated with or without IFN-γ. *n* = 2 biologically independent experiments (**d**), *n* = 3 biologically independent experiments (**e**). **f** Flow cytometry analysis of PD-L1 levels on the surface of control and OTUB2-OE LL/2 cells treated with or without IFN-γ. *n* = 3 biologically independent experiments. **g** Screening of other five DUBs that regulate PD-L1 protein level. NCI-H358 cells individually were infected with shRNAs targeting different DUBs, and the surface expression of PD-L1 was detected. Each value above the bar is expressed as a fold change relative to the control. *n* = 3 biologically independent experiments. **h** Western blot analysis of PD-L1 in control and OTUB2-KD NCI-H358 cells treated with or without IFN-γ. **i** Western blot analysis of PD-L1 in NCI-H358 cells overexpressing a control vector (EV), OTUB2 (WT) or the OTUB2 C51S mutant (C51S). **j** Western blot analysis of PD-L1 expression in vector control and OTUB2-KD NCI-H358 cells treated with cycloheximide (CHX) at the indicated time points. **k** Quantitative estimates of PD-L1 levels based on western blot analysis of cells treated with CHX. **l** Western blot analysis of PD-L1 expression in vector control and OTUB2-OE 293 T cells treated with CHX at the indicated time points. **m** Quantitative estimates of PD-L1 levels based on western blot analysis of cells treated with CHX. The experiments were repeated three times independently with similar results. The data are presented as the mean ± s.e.m., and *P* values were calculated using an unpaired two-sided Student's *t* test (**a**–**f**).

protein in OTUB2-KD tumor cells (Fig. 3j, k) and slower degradation of PD-L1 protein in OTUB2-overexpressing cells (Fig. 3l, m). Taken together, these data identify OTUB2 as a specific positive regulator of PD-L1 in different tumor cells.

## OTUB2 stabilizes PD-L1 by Lys-48-linked polydeubiquitination

To determine whether OTUB2-mediated PD-L1 stabilization is regulated by deubiquitination, we asked whether OTUB2 deubiquitinates PD-L1 to increase protein stability. We analyzed PD-L1 ubiquitination in the presence of the proteasome inhibitor MG132 and found that MG132-induced PD-L1 ubiquitination was abolished by OTUB2 OE (Fig. 4a). In contrast, MG132-induced PD-L1 polyubiquitination was enhanced by OTUB2 KO (Fig. 4b). Furthermore, reintroducing sgRNA-resistant OTUB2 into OTUB2-KO 293 T cells rescued their ability to reduce the polyubiquitinated PD-L1 level (Fig. 4c and Supplementary Fig. 12a). We also confirmed the effect of depleted OTUB2 on the level of endogenous PD-L1 ubiquitination (Fig. 4d and Supplementary Fig. 12b). We further asked whether OTUB2 can induce the deubiquitination of non-glycosylated (premature or misfolded) PD-L1 protein[26]. We found that OTUB2 suppressed polyubiquitination of both non-glycosylated and glycosylated PD-L1 (Fig. 4e). Because protein degradation is typically associated with Ub K48 or K63 chain polyubiquitination[41,42] and previous evidence has shown that OTUB2 cleaves Lys48 and Lys63 chains preferentially[43,44], we next tested whether OTUB2 could remove K48- or K63-linked polyubiquitin chains on PD-L1. We found that OTUB2 reduced the endogenous K48-linked polyubiquitination of PD-L1 but had no effect on K63-linked polyubiquitination (Fig. 4f, g). To determine whether the protein-protective effects depended on the catalytic activity of OTUB2, sgRNA-resistant OTUB2-WT and OTUB2-C51S were individually transfected into OTUB2-KO 293 T cells. Only reintroducing OTUB2-WT reduced MG132-induced PD-L1 polyubiquitination in OTUB2-KO cells, suggesting that OTUB2 regulated PD-L1 protein stability through its deubiquitination-associated enzymatic activity (Fig. 4h). Moreover, an in vitro deubiquitination assay further confirmed that OTUB2 was able to remove K48-linked polyubiquitin chains from PD-L1 (Fig. 4i).

To determine whether OTUB2 affects PD-L1 degradation through the proteasome, we examined the effect of MG132 or the lysosomal inhibitor chloroquine on the PD-L1 levels in OTUB2-KD lung cancer cells. Reduced PD-L1 levels in OTUB2-deficient cells could be restored only by incubation with MG132 (Fig. 5a, b and Supplementary Fig. 13a, b). As OTUB2 interacts with the ICD of PD-L1, we speculated that OTUB2 can remove polyubiquitin chains from the ICD of PD-L1. Based on homology analysis of the amino acids of PD-L1 across different species, we identified three conserved lysine residues located within the ICD that might be deubiquitination sites for OTUB2 (Fig. 5c). We constructed two mutants, PD-L1ΔICD (devoid of the ICD) and PD-L1 3KR (three lysine residues were mutated to arginine), and examined whether OTUB2 could still reduce the polyubiquitination of PD-L1. Ectopically expressed PD-L1 3KR but not PD-L1ΔICD associated with

OTUB2 in HEK293T cells (Figs. 2f, 5d). In line with the lack of interaction between OTUB2 and PD-L1ΔICD, we observed that compared with full-length PD-L1 (PD-L1 FL), PD-L1ΔICD was barely modified by polyubiquitination, and OTUB2 did not reduce the polyubiquitination of PD-L1ΔICD (Fig. 5e). Furthermore, we found that the polyubiquitination of PD-L1 FL was significantly increased while PD-L1 3KR displayed significantly less polyubiquitination (Fig. 5f). These results indicate that OTUB2 stabilizes PD-L1 by directly cleaving K48-linked polyubiquitin chains in the ICD of PD-L1 and preventing proteasomal degradation of PD-L1.

## OTUB2 mediates ER-dependent degradation of PD-L1

To further explore the cellular location at which OTUB2 prevents the proteasomal degradation of PD-L1, we performed immunofluorescence (IF) following Duolink in situ PLAs using antibodies targeting different organelles to confirm the subcellular location of the OTUB2–PD-L1 interaction. OTUB2 proteins were found to be mainly located in the nucleus and cytosol, while PD-L1 proteins were found to be mainly located in the membrane and cytosol (Fig. 6a and Supplementary Fig. 14a, b). Significantly reduced PD-L1 staining was observed in OTUB2-KO tumor cells (Supplementary Fig. 14a). We found that the interaction between OTUB2 and PD-L1 (orange signals) mainly colocalized with the ER marker calnexin (Fig. 6b). The PLA showed that PLA signals representing the OTUB2-PD-L1 interaction were mainly localized in the ER (Fig. 6b). Furthermore, we found that the addition of eeyarestatin I (Eer I), a specific inhibitor of ERAD, rescued PD-L1 levels in both OTUB2-KD human tumor cells and OTUB2-KO mouse tumor cells (Fig. 6c and Supplementary Fig. 15a, b). We also found that after MG132 treatment, PD-L1 accumulated in the cytoplasm and mainly colocalized with the ER marker calnexin (Fig. 6d). These data suggest that OTUB2 regulates PD-L1 in the ER. Moreover, we compared PD-L1 and OTUB2 expression in a panel of human tumor cell lines, and found that PD-L1 expression was positively associated with OTUB2 abundance (Fig. 6e, f). Furthermore, reintroducing sgRNA-resistant OTUB2 but not the C51S mutant into OTUB2-deficient cells rescued PD-L1 expression (Fig. 6g and Supplementary Fig. 15c), thereby ruling out the possibility that off-target OTUB2 KD or KO might be responsible for the altered phenotypes.

## Depletion of OTUB2 promotes CD8⁺ CTL-mediated killing of tumor cells and enhances intratumoral T-cell infiltration

To evaluate whether the altered PD-L1 levels after OTUB2 depletion can affect antitumor immune response in the TME, we quantified effector immune cells in control and OTUB2-KO mouse tumors by flow cytometry and IF staining. The flow cytometry analyses demonstrated significant increases in the numbers of intratumoral CD45⁺ immune cells and CD8⁺ CTLs in OTUB2-KO B16F10 tumors (Fig. 7a, b and Supplementary Fig. 16a). We also observed that the percentages of IFN-γ⁺ and granzyme B⁺ (GZMB⁺) CD8⁺ CTLs were significantly increased in OTUB2-KO tumors (Fig. 7c, d). IF staining indicated that depletion of

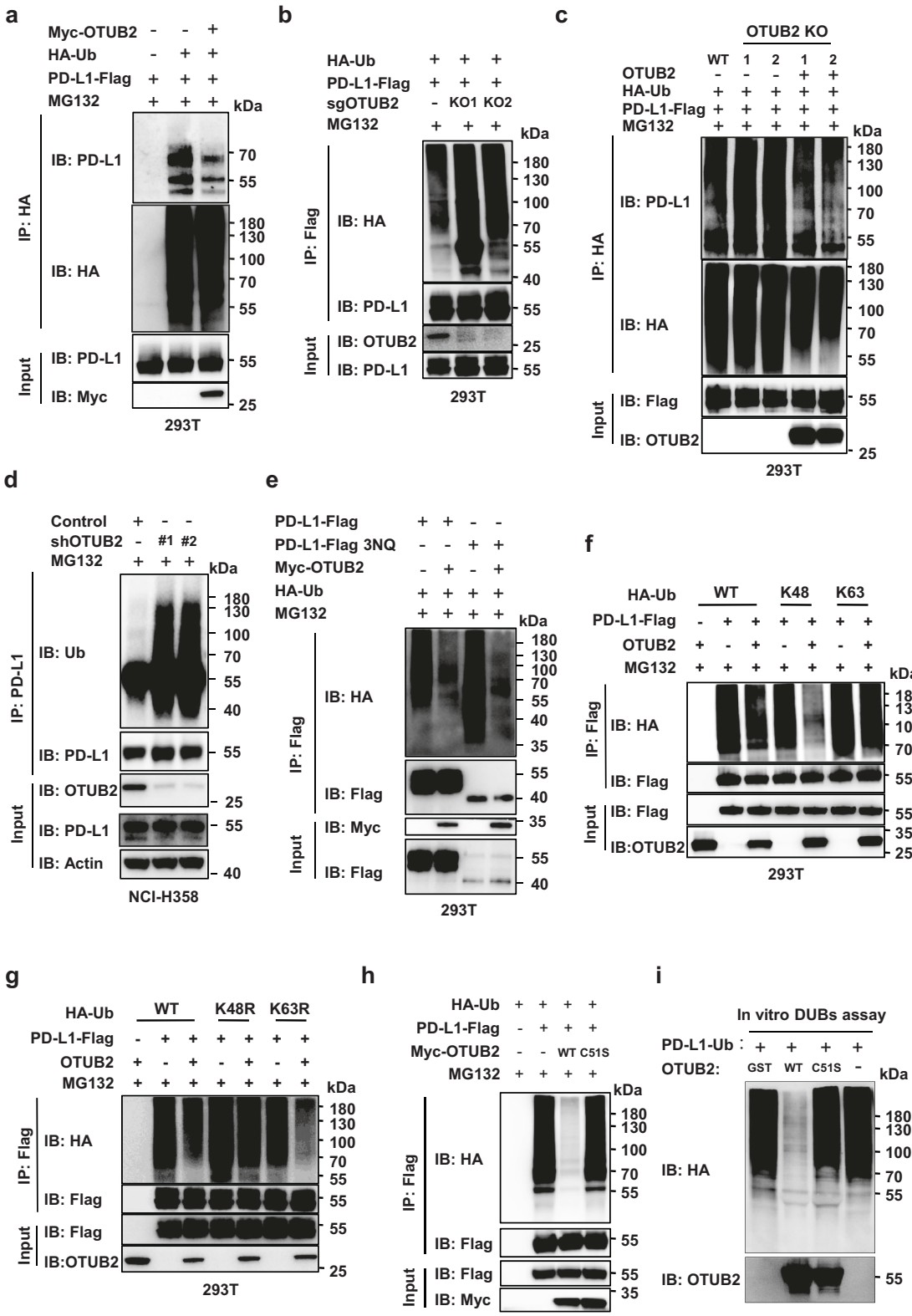

OTUB2 caused an overall increase in the intratumoral infiltration of CD8[+] cells and GZMB[+] CTLs and an overall decrease in the intratumoral infiltration of PD-L1[+] cells in OTUB2-KO LL/2 tumors (Fig. 7e).

Given that PD-L1 upregulation can lead to immunosuppression via inhibition of CTLs, we then determined the effect of OTUB2 KO on the CTL-mediated killing of tumor cells by using an OT-1 transgenic mouse model. Our results indicated that OTUB2-KO tumor cells were

significantly more susceptible to killing by OT-1 CTLs than were control cells (Fig. 7f and Supplementary Fig. 16b). We then performed T-cell proliferation assays to further assess whether OTUB2 KO influences the function of PD-L1 in T-cell activation and proliferation. We found that the number of activated T cells was significantly increased in a coculture of OT-1 T cells and OTUB2-KO LL/2 tumor cells compared with a coculture of OT-1 cells and control tumor cells (Fig. 7g).

**Fig. 4 | OTUB2 maintains PD-L1 protein stability by cleaving K48-linked poly-ubiquitin chains from PD-L1. a** Ubiquitination assay assessing PD-L1 in 293 T cells cotransfected with HA-Ubiquitin (Ub), PD-L1-Flag and Myc-OTUB2 and treated with MG132. **b** Ubiquitination assay assessing PD-L1 in OTUB2-KO 293 T cells cotransfected with HA-Ub and PD-L1-Flag and treated with MG132. **c** Determination of PD-L1 polyubiquitination levels in OTUB2-KO 293 T cells after re-expression of OTUB2-WT. Immunoprecipitation was performed with an anti-HA antibody. **d** Determination of endogenous PD-L1 polyubiquitination levels in OTUB2-KD NCI-H358 cells. Immunoprecipitation was performed with an anti-PD-L1 antibody. **e** OTUB2 was introduced to analyze the ubiquitination of glycosylated and non-glycosylated PD-L1 in 293 T cells. 3NQ represents substitution of each of the three asparagine (N) to glutamine (Q)− N192Q, N200Q and N219Q, which is critical for PD-L1 glycosylation.

**f** The influence of OTUB2 on K48-linked or K63-linked ubiquitinated PD-L1 was determined by western blot analysis. The PD-L1-Flag ubiquitination levels in 293 T cells transfected with HA-Ub mutants (WT, K48only and K63only), PD-L1-Flag and Myc-OTUB2 were analyzed by a Flag pull-down assay. **g** Detection of non-K48-linked or non-K63-linked polyubiquitination of PD-L1 affected by OTUB2. The PD-L1-Flag ubiquitination levels in 293 T cells transfected with HA-Ub mutants (WT, K48R and K63R), PD-L1-Flag and Myc-OTUB2 were analyzed by a Flag pull-down assay. **h** Examination of the effects of WT OTUB2 or the C51S mutant on the deubiquitination of PD-L1. **i**, In vitro deubiquitination assay assessing ubiquitinated PD-L1-Flag with purified GST-OTUB2 or GST-OTUB2-C51S. The experiments were repeated three times independently with similar results. Source data are provided as a Source Data file.

Furthermore, we also performed T-cell cytotoxicity assays to confirm that the immune-inhibitory function of PD-L1 was affected by OTUB2 in human tumor cells, OTUB2 KD rendered tumor cells more sensitive to human peripheral blood mononuclear cell (PBMC)-mediated killing (Fig. 7h and Supplementary Fig. 16c). These results indicate that depletion of OTUB2 enhances antitumor immunity by regulating PD-L1 abundance and activating CTLs.

## OTUB2 expression is associated with higher PD-L1 and lower CD8⁺ CTL levels in NSCLC patient samples

Inspired by the previously observed association among elevated OTUB2 mRNA levels, worse overall or disease-free survival, and reduced CD8⁺ T-cell infiltration (estimated from a CD8⁺ T-cell signature) in a pancancer analysis of 32 human tumor datasets from the TCGA, we further investigated the clinical relevance of OTUB2 protein expression in cohorts of lung squamous cell carcinoma (LUSC) patients and lung adenocarcinoma (LUAD) patients. We first quantitatively assessed OTUB2 protein expression in human LUSC specimens by immunohistochemistry (IHC) and compared the differential expression of OTUB2 between cancer tissues and adjacent para-cancerous tissues. The IHC results revealed that OTUB2 was more highly expressed in cancer tissues than in adjacent paracancerous tissues (Fig. 8a, b). In paired patients, OTUB2 was also upregulated in the majority of cancer tissues compared to adjacent paracancerous tissues (Fig. 8c). Consistently, analysis of the LUSC cohort profiled by the TCGA also revealed that OTUB2 was more highly expressed in cancer tissues than in normal tissues (Supplementary Fig. 17a), while the CTL signature genes CD8A and GZMB were expressed at significantly lower levels in cancer tissues than in normal tissues (Supplementary Fig. 17b, c). We also performed overall survival analysis of LUSC patient samples and found that LUSC patients with high levels of OTUB2 exhibited significantly poorer overall survival (Fig. 8d and Supplementary Fig. 17d). Additionally, IHC analysis of the LUAD patient samples also revealed that OTUB2 was more highly expressed in cancer tissues than in normal tissues (Supplementary Fig. 18a–d). A similar survival result was observed in LUAD patients (Supplementary Fig. 18e).

We next assessed the correlations between OTUB2 and PD-L1 protein expression or OTUB2 and CD8 protein expression in human LUSC specimens. We found that OTUB2 protein levels were positively correlated with PD-L1 expression (Pearson $r = 0.5443$, $p < 0.0001$) (Fig. 8e). Representative images of high and low staining for OTUB2 and PD-L1 are shown in Fig. 8f. We also found that OTUB2 protein levels were negatively correlated with CD8 expression (Pearson $r = -0.3156$, $p < 0.0037$) (Fig. 8g). Representative images of high and low staining for OTUB2 and CD8 are shown in Fig. 8h. We also found that OTUB2 protein levels were positively correlated with PD-L1 expression (Pearson $r = 0.5922$, $p < 0.0001$) in human LUAD specimens (Supplementary Fig. 18f). Representative images of high and low staining for OTUB2 and PD-L1 are shown in Supplementary Fig. 18g. These findings suggest that OTUB2 has potential clinical significance and could serve as a biomarker for cancer diagnosis.

## A self-developed OTUB2 inhibitor can specifically reduce PD-L1 levels in cells and promote antitumor immunity

Given the important role of OTUB2-mediated PD-L1 stability in suppression of tumor growth, we reasoned that targeting OTUB2 could be beneficial in the treatment of cancer. To preclinically prove this concept, we aimed to identify inhibitors of OTUB2. By virtual screening of chemical libraries consisting of 494,400 small-molecule drugs, the chemical compounds capable of binding to the catalytic pocket of OTUB2 were scored. The top 20 hits were selected (Supplementary Table 2) and their ability to inhibit the ubiquitylation of PD-L1 in vitro was tested. Notably, OTUB2-IN-1, a compound of previously unknown function, was identified as a potential candidate to bind the catalytic pocket of OTUB2 (Fig. 9a). Four amino acid residues (Thr45, Gly49, Ser223 and His224) in the cavity center, were projected to form hydrogen bonds with two carboxyl groups of OTUB2-IN-1 (Fig. 9b and Supplementary Fig. 19a). Incubation of recombinant GST-OTUB2 protein with OTUB2-IN-1 resulted in a significant inhibition of Ub-R110 cleavage by GST-OTUB2 in a dose-dependent manner (Supplementary Fig. 19b−d), suggesting that OTUB2-IN-1 can inhibit the hydrolytic activity of recombinant GST-OTUB2 protein toward its substrate Ub-R110 in vitro. OTUB2-IN-1 increased ubiquitinated PD-L1 levels in a dose-dependent manner, as evidenced by in vitro deubiquitylation assay (Fig. 9c, d and Supplementary Fig. 19e). Surface plasmon resonance (SPR) assays gave a dissociation constant ($K_D$) value of approximately 12 μM for OTUB2-IN-1 binding to OTUB2 (Fig. 9e). We next examined the effect of OTUB2-IN-1 on PD-L1 degradation, and found that OTUB2-IN-1 was able to reduce the PD-L1 level in tumor cells in a dose-dependent manner, but it failed to affect the stability of OTUB2 (Fig. 9f and Supplementary Fig. 20a−c). To ensure that the ability of OTUB2-IN-1 to reduce PD-L1 expression was due to the impaired deubiquitinase activity of OTUB2 rather than the impaired binding of OTUB2 to PD-L1, we further verified the ability of OTUB2-IN-1 to block the binding of OTUB2 to PD-L1 using a blocking chemiluminescence immunoassay (CLIA) and a co-IP assay. We found that OTUB2-IN-1 did not interfere with protein interactions between OTUB2 and PD-L1 (Fig. 9g and Supplementary Fig. 21). Furthermore, 10 μM OTUB2-IN-1 exhibited no obvious inhibitory effect on the viability of tumor cells (Fig. 9h and Supplementary Fig. 22a, b). We further evaluated the therapeutic potential of OTUB2-IN-1 using a mouse LL/2 tumor model (Fig. 9i). OTUB2-IN-1 treatment significantly suppressed tumor growth and prolonged mouse survival, and no obvious toxicity was observed (Fig. 9j and Supplementary Fig. 23a, b). To clarify the effect of OTUB2-IN-1 in tumor inhibition through regulation of PD-L1 in vivo, LL/2 cells with stably expressed PD-L1 and control plasmids were used to establish tumor models (Supplementary Fig. 24a). As expected, in comparison to the control group, overexpressed PD-L1 resulted in a substantial increase of tumor growth, whereas OTUB2-IN-1 significantly reduced the PD-L1-overexpressing tumor growth (Fig. 9k and Supplementary Fig. 24b, c). We also found that the antitumor effect of OTUB2-IN-1 could be affected by PD-L1 overexpression and no obvious toxicity was observed (Fig. 9k and Supplementary Fig. 24b−d). This rescue experiment indicates that OTUB2-IN-1 attenuates tumor

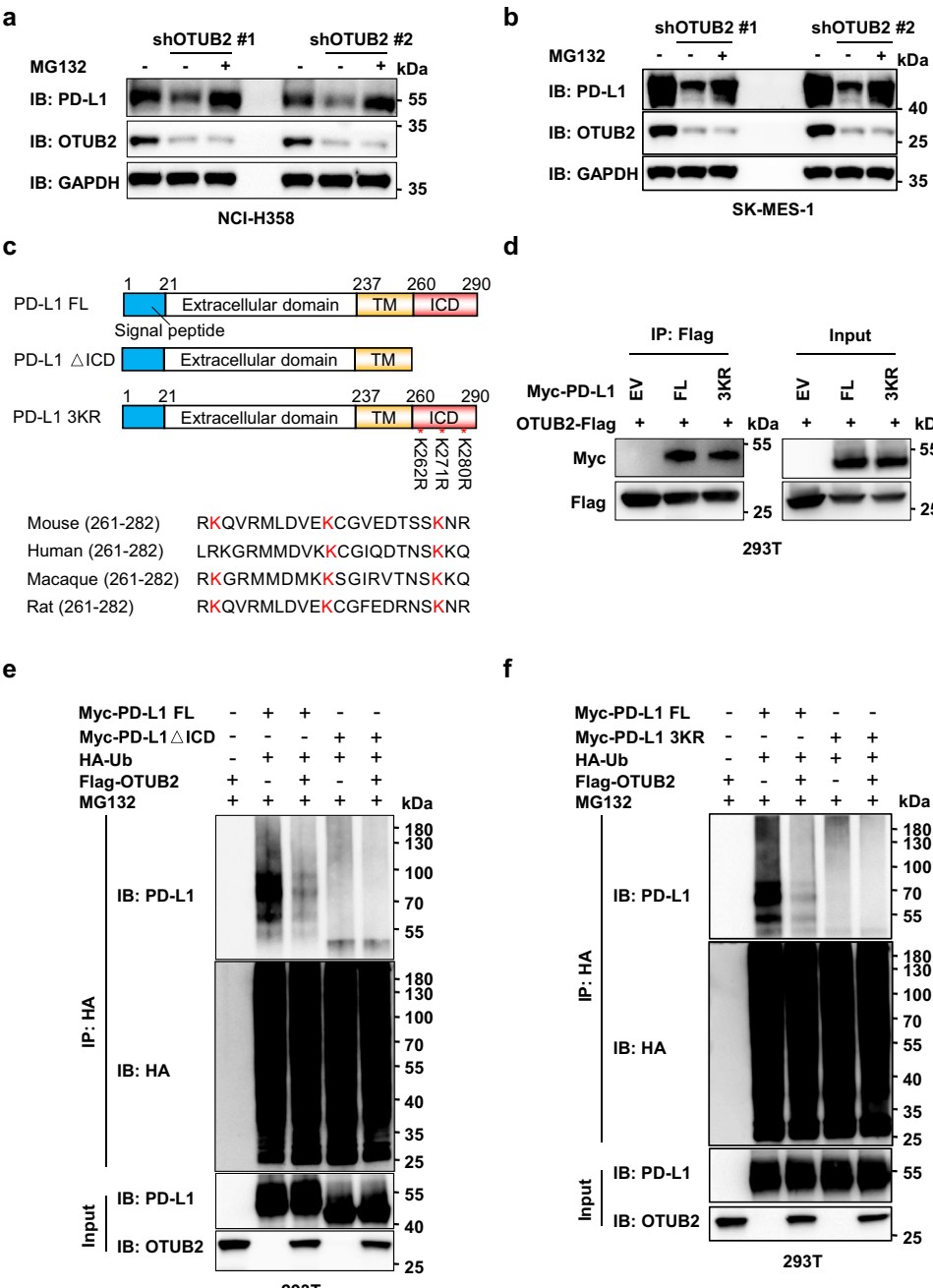

**Fig. 5 | OTUB2 controls the endoplasmic reticulum (ER)-mediated degradation of PD-L1 in tumor cells. a** NCI-H358 tumor cells with or without OTUB2 KD in the absence or presence of 10 μM MG132 were collected to analyze PD-L1 expression. **b** SK-MES-1 tumor cells with or without OTUB2 KD in the absence or presence of 10 μM MG132 were collected to analyze PD-L1 expression. **c** Schematic representation of PD-L1 FL, △ICD and 3KR constructs. Below: sequence alignment of the conserved lysine residues in the ICD region of the PD-L1 orthologs of different species. **d** The interaction between OTUB2-Flag and Myc-PD-L1 ΔICD or 3KR was detected by a co-IP assay. **e** The influence of the ICD of PD-L1 on the polyubiquitination of PD-L1 was determined by western blot analysis. The Myc-PD-L1 FL or Myc-PD-L1-△ICD ubiquitination levels in 293 T cells transfected with HA-Ub, Myc-PD-L1 mutants and Flag-OTUB2 were analyzed by an HA pull-down assay. **f** The influence of the three key lysine residues in the ICD of PD-L1 on the poly-ubiquitination of PD-L1 was determined by western blot analysis. The Myc-PD-L1 FL or Myc-PD-L1-3KR ubiquitination levels in 293 T cells transfected with HA-Ub, Myc-PD-L1 mutants and Flag-OTUB2 were analyzed by an HA pull-down assay. The experiments were repeated three times independently with similar results. Source data are provided as a Source Data file.

growth through regulation of PD-L1. The therapeutic potential of OTUB2-IN-1 was also observed in mouse B16-F10 and KLN205 tumor models (Supplementary Fig. 25a–d). Furthermore, IHC analyses of tumors derived from OTUB2-IN-1-treated mice showed significant intratumoral infiltration of CD8+ and GZMB+ CTLs and dramatically reduced expression of PD-L1 compared to those derived from untreated mice (Fig. 9l–n and Supplementary Fig. 26a–d). Since

OTUB2 can promote cancer progression by regulating Hippo, NF-kB and Akt/mTOR pathways[28–30], we next investigated the effects of OTUB2-IN-1 on regulating Hippo, NF-κB and Akt/mTOR pathways. In LL/2 tumors, OTUB2-IN-1 treatment significantly reduced the expression of YAP and phosphorylated p65, but had no effect on the expression of phosphorylated Akt (Fig. 9o, p). Interestingly, OTUB2-IN-1 treatment only significantly reduced the expression of

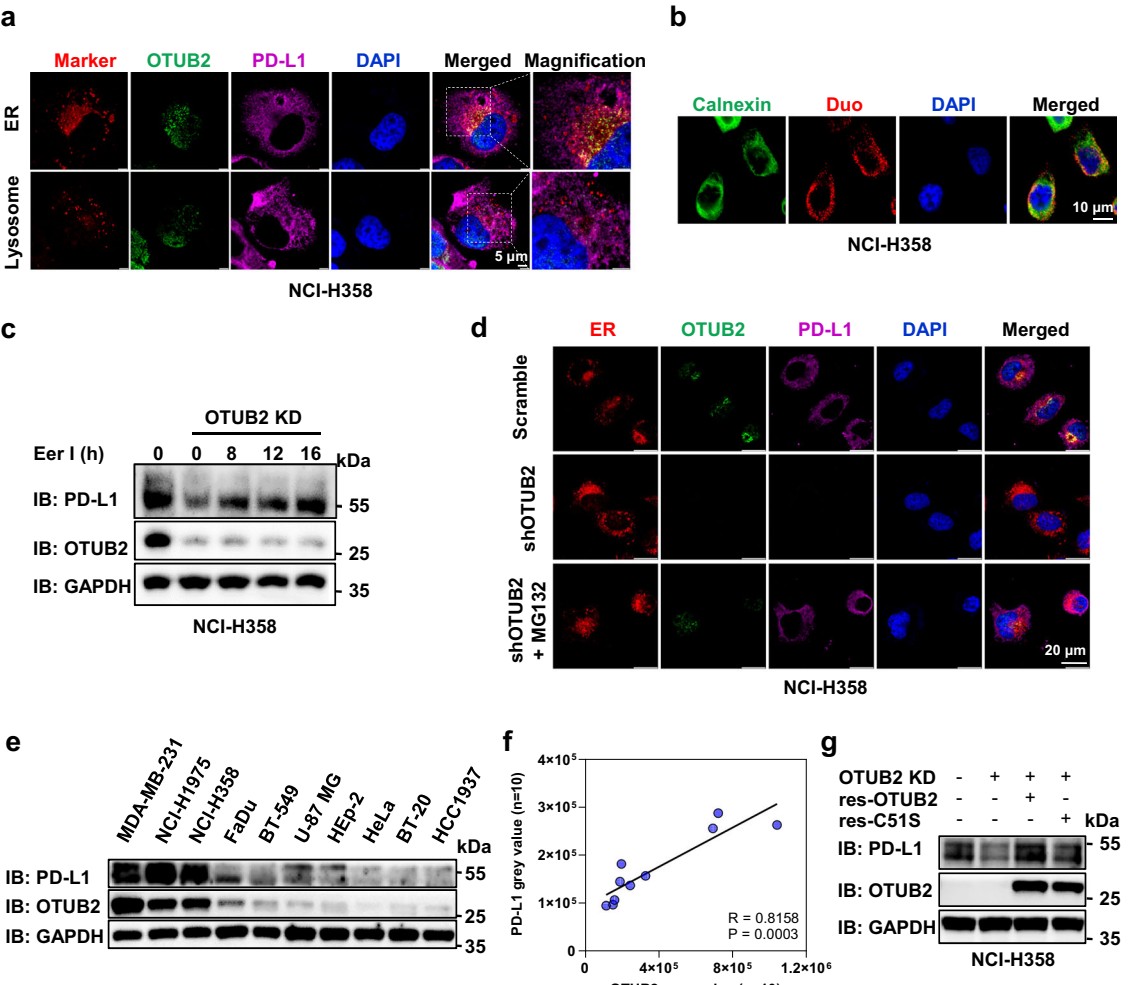

**Fig. 6 | OTUB2 disrupts the degradation of PD-L1 in the ER. a** NCI-H358 tumor cells were immunostained with an anti-PD-L1 antibody (purple), an anti-OTUB2 antibody (green), DAPI (blue) and lyso- or ER-specific trackers (red). Scale bar, 5 μm. **b** The subcellular colocalization of PD-L1 and OTUB2 was detected by a Duolink in situ PLA and ER-specific trackers. Scale bar, 10 μm. **c** NCI-H358 tumor cells with or without OTUB2 KD in the absence or presence of 20 μM Eer I were collected to analyze PD-L1 expression at the indicated times. **d** NCI-H358 tumor cells with or without OTUB2 KD in the presence of 10 μM MG132 were immunostained with an anti-OTUB2 antibody (green), an anti-PD-L1 antibody (purple), DAPI (blue) and ER-specific trackers (red). Scale bar, 20 μm. **e** Western blot analysis of PD-L1 and OTUB2 expression in the indicated human tumor cell lines. **f** Pearson correlation analysis to determine the degree of association between OTUB2 and PD-L1 by immunoblotting. **g** Determination of the PD-L1 levels in OTUB2-KD NCI-H358 cells after re-expression of OTUB2-WT or OTUB2-C51S. The experiments were repeated three times independently with similar results. The data in (**f**) are presented as the mean ± s.e.m., and *P* values were calculated using Spearman's correlation test.

phosphorylated Akt in B16-F10 tumors (Supplementary Fig. 27a, b) and significantly reduced the expression of phosphorylated p65 in KLN205 tumors (Supplementary Fig. 28a, b), suggesting that OTUB2 may possess more general roles in regulation of PD-L1 in various cancer types. Together, these results indicate that degradation of PD-L1 by OTUB2-IN-1 is responsible for its immune effect on growth inhibition.

## Discussion
Our in vitro and in vivo data demonstrate that OTUB2 functions as a negative regulator of T-lymphocyte-mediated antitumor immunity by modulating PD-L1 stabilization on tumor cells and thereby promotes tumor immune evasion. PD-L1 protein homeostasis is critically important in a variety of biological processes, disruption of which can lead to immune-related human diseases[45]. We show that PD-L1 protein homeostasis is tightly regulated by the DUB OTUB2. OTUB2 can interact with PD-L1 and deubiquitinate and stabilize the PD-L1 protein by intervening in the ERAD pathway. Interestingly, we found that Cys51, the key residue of OTUB2 enzymatic activity, is also essential for its interaction with PD-L1, suggesting that OTUB2 affects PD-L1 not only relying on its enzymatic activity but also on its interaction with

PD-L1. This is similar with a previous study showing that OTUB2 C51S loses its binding ability to STAT1 and fails to deubiquitinate STAT1[46]. However, several reports have described different modes of action of DUBs in regulating their substrates, such as deubiquitination of YAP/TAZ by OTUB2[29] and deubiquitination of PD-L1 by CSN5[23], the key residues responsible for the enzymatic activity of DUBs are not essential for DUBs−substrate interactions. The ERAD pathway not only uses to clear misfolded proteins from the ER for cytosolic proteasomal degradation but also maintains appropriate levels of membrane-associated proteins[47]. In this study, we demonstrate that OTUB2 is likely to regulate the physiological quantity of PD-L1 proteins via reducing polyubiquitination of both premature and mature PD-L1. Of note, our results show that OTUB2 is a critical DUB that inhibits PD-L1 proteasomal degradation independent of IFN-γ stimulation in multiple cancers. In fact, PD-L1 protein stability is maintained at multiple levels of complex regulation, including specific E3 ligases, DUBs, proteases, glycosylases and regulators of the proteasome and lysosomes[48]. Previous studies demonstrated that USP7, USP9X and USP22 stabilize PD-L1 in gastric cancer, oral squamous cell carcinoma or liver cancer, respectively[38–40]. The regulation of PD-L1 degradation by CSN5 or

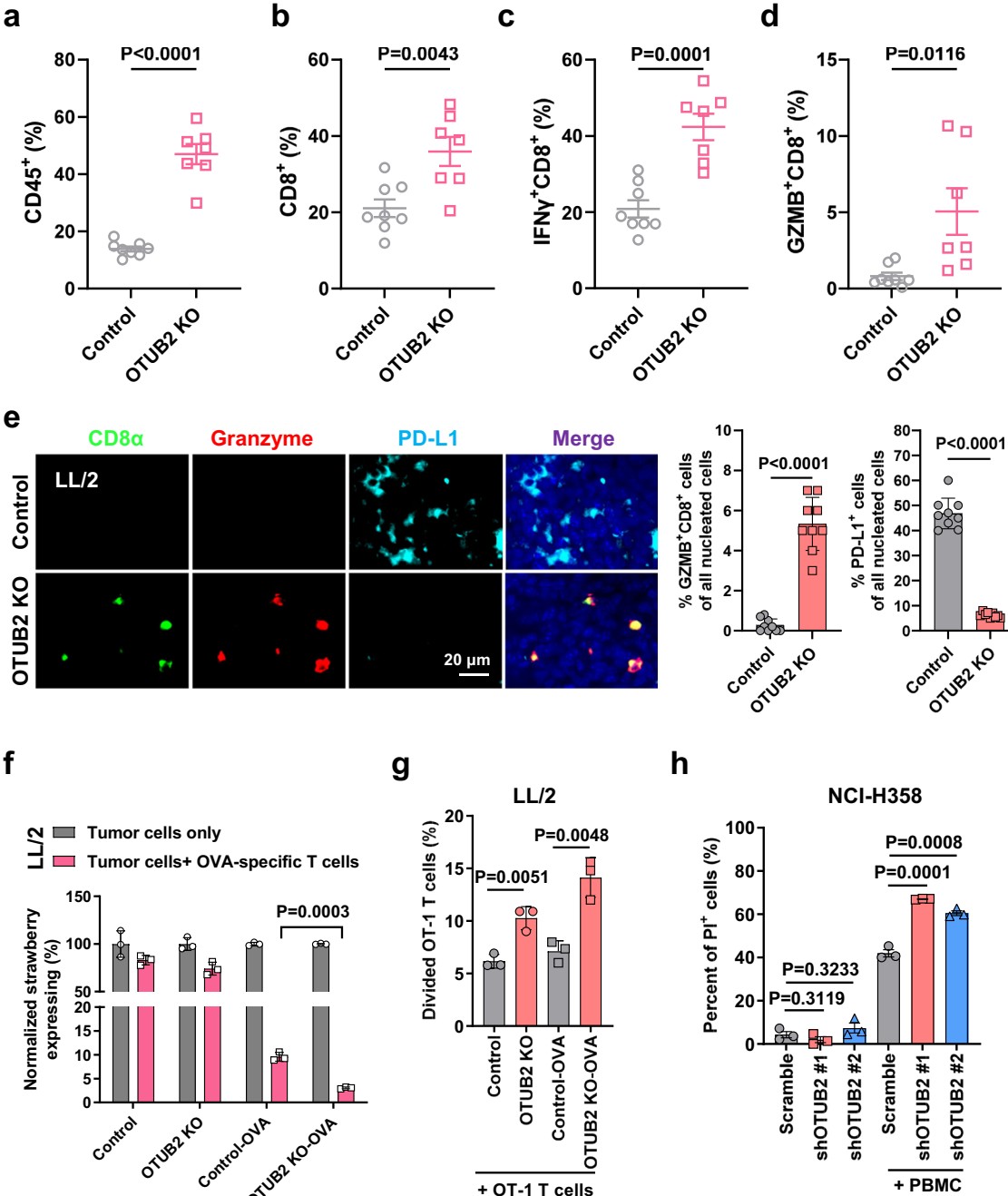

**Fig. 7 | Depletion of OTUB2 enhances tumor cell sensitivity to CD8[+] T-cell-mediated cytotoxicity and intratumoral T-cell infiltration. a–d** Quantitative estimation of the percentages of various immune effector cells of tumor tissue in vector control and OTUB2-KO B16-F10 tumors, as determined by flow cytometry. $n = 8$ tumors in control group, $n = 7$ tumors in OTUB2 KO group. **e** Representative images of IHC staining with DAPI, anti-CD8α, anti-GZMB and anti-PD-L1 in vector control or OTUB2-KO LL/2 tumors grown in syngeneic C57BL/6 mice. Scale bars, 20 μm. Three fluorescent fields for each of the three samples were counted. $n = 9$ biologically independent samples. **f** Quantitative estimation of the fractions of live vector control and OTUB2-KO LL/2-Strawberry tumor cells remaining after 24 h of incubation with activated OVA-specific T cells. $n = 3$ biologically independent experiments. **g** Quantification of the numbers of live activated CD8[+] OVA-specific T cells after 48 h of coculture with vector control or OTUB2-KO LL/2 tumor cells. $n = 3$ biologically independent experiments. **h** Flow cytometry analysis of the peripheral blood mononuclear cell (PBMC)-mediated killing of vector control or OTUB2-KD NCI-H358 tumor cells using propidium iodide (PI) staining. $n = 3$ biologically independent experiments. The above experiments were repeated three times independently with similar results. The data are presented as the mean ± s.e.m., and $P$ values were calculated using an unpaired two-sided Student's $t$ test (**a**–**h**).

OTUB1 is mainly found in breast cancer[23,37]. In this study, we demonstrate OTUB2-mediated PD-L1 regulation as a common feature in various human and mouse cancers, particularly in NSCLC. We found OTUB2 to be the top positive regulator of PD-L1 protein stability in NSCLC among all tested DUBs, since OTUB2 depletion showed the most significant impact on PD-L1 abundance. However, other five DUBs

function differently in regulating PD-L1 abundance in NSCLC cells. Knockdown of USP22 or OTUB1 only significantly reduces PD-L1 abundance in NCI-H358 cells but not in SK-MES-1 cells. Meanwhile, the PD-L1 protein level remains unchanged or slightly changed after knockdown of CSN5, USP9X or USP7. How OTUB2 functions differently from other DUBs in regulating PD-L1 abundance in other cancers

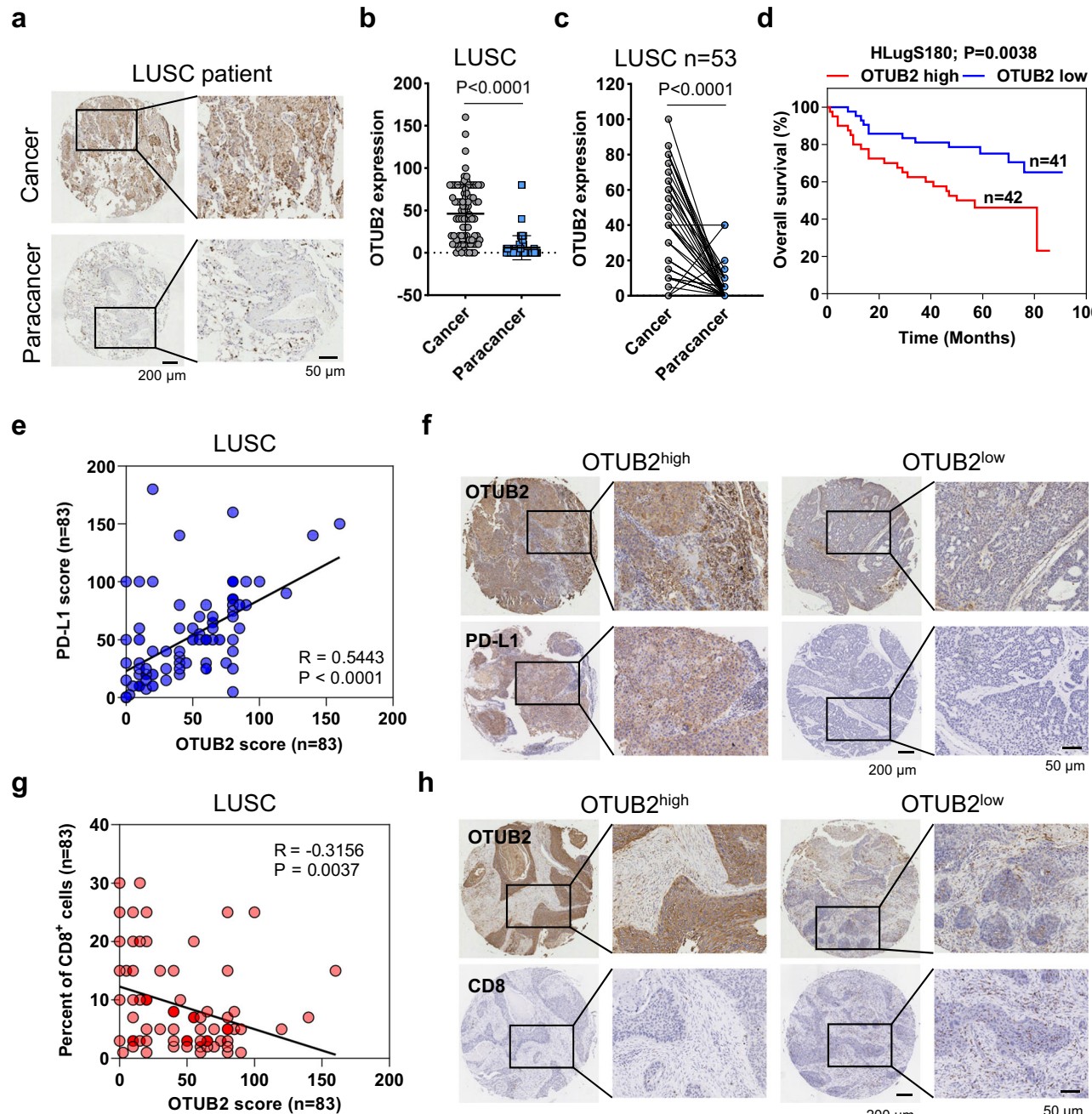

**Fig. 8 | The expression of OTUB2 is positively correlated with PD-L1 abundance or the density of CD8+ T cells in LUSC patient samples. a** Representative images of IHC staining for OTUB2 in cancer tissues and paracancerous tissues from 165 LUSC patients. Scale bars, 200 μm or 50 μm. **b** Comparison of OTUB2 expression between the cancer tissues ($n = 83$) and paracancerous tissues ($n = 82$) of LUSC patients. **c** OTUB2 expression in the paired tissue samples of 53 LUSC patients. **d** Kaplan–Meier curve for the overall survival of LUSC patients based on OTUB2 expression scores calculated by IHC staining. Patients were divided into a high OTUB2 group ($n = 42$) or low OTUB1 group ($n = 41$). **e** Pearson correlation analysis to determine the association between OTUB2 and PD-L1 by IHC staining scores ($n = 83$). **f** Representative images of IHC staining for PD-L1 in LUSC cancer tissues with high or low OTUB2 expression. Scale bars, 200 μm or 50 μm. **g** Pearson correlation analysis to determine the association between OTUB2 and CD8α by IHC staining scores ($n = 83$). **h** Representative images of IHC staining for CD8α in LUSC cancer tissues with high or low OTUB2 expression. Scale bars, 200 μm or 50 μm. The data are presented as the mean ± s.e.m., and *P* values were calculated by Spearman's correlation test (**e, g**) except for in (**d**), in which *P* values were calculated by the log-rank test, and (**b, c**), in which *P* values were calculated by an unpaired or paired two-sided Student's *t* test.

needs to be further studied in the future. Moreover, we demonstrate that OTUB2 is a general regulator of PD-L1 since genetic depletion or pharmaceutical inhibition of OTUB2 expression profoundly impacts PD-L1 protein stability in multiple cancers, especially in tumors generally expressing low levels of PD-L1. Our findings reveal an underlying molecular mechanism of PD-L1 posttranslational modification, which represents a conceivable tumor-targeting strategy.

We also demonstrated that PD-L1 downregulation by depletion of OTUB2 resulted in inhibition of tumor growth by promoting robust intratumoral infiltration of CTLs. Notably, emerging evidence indicates that selective inhibition of DUBs may represent an effective approach for cancer treatment[34,49]. Before this study, OTUB2 had not been targeted successfully with effective therapeutic strategies, limiting the immediate translational value of targeting OTUB2. However, several

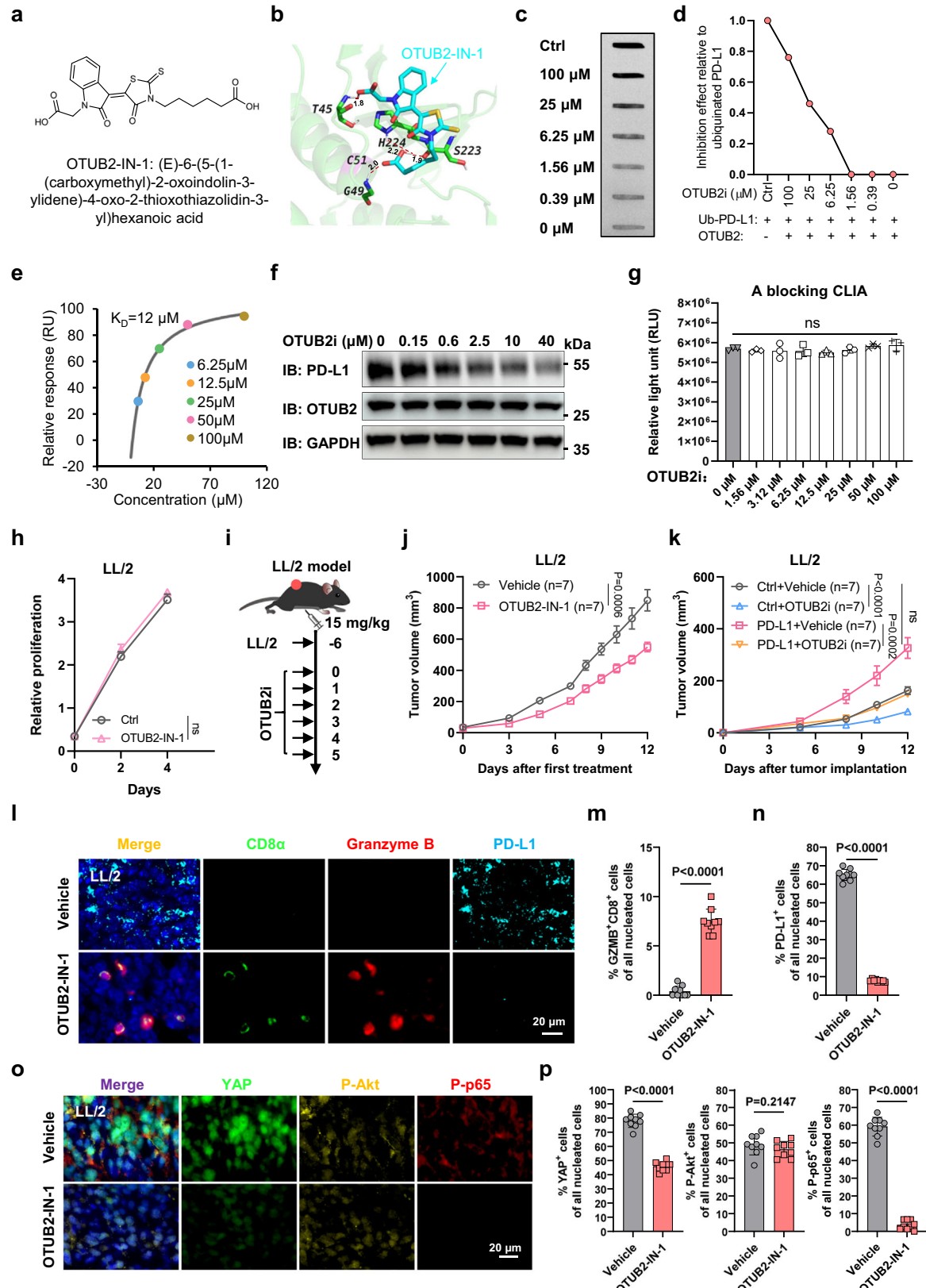

studies have developed several small-molecule inhibitors for multiple other DUBs, such as JOSD1[50], USP7[51], USP10 and USP13[52]. In this study, we presented preclinical evidence, providing proof-of-concept data indicating that targeting OTUB2 with a specific inhibitor, OTUB2-IN-1, is an effective therapeutic strategy for the treatment of cancers. We demonstrated that OTUB2-IN-1 reduced PD-L1 protein turnover through inhibition of OTUB2 catalytic activity rather than blocking the OTUB2-PD-L1 interactions and exhibited strong growth-suppressive effects on multiple cancers by attenuating immunosuppression. Considering that the preclinical evaluation of OTUB2-IN-1 in this study is preliminary, the systemic effects and antitumor activity of OTUB2-IN-1 in more clinically relevant animal models should be carefully evaluated in future studies.

**Fig. 9 | Inhibition of OTUB2 with a selective compound suppresses tumor growth by promoting antitumor immunity. a** The chemical structure of OTUB2-IN-1. **b** A schematic diagram of the docking interaction between the indicated residues in the catalytic pocket of OTUB2 and OTUB2-IN-1. OTUB2-IN-1 is labeled in cyan. The hydrogen bonds are indicated by red dashed lines. Dot blot analysis (**c**) and quantitative estimates (**d**) of the effect of OTUB2-IN-1 (OTUB2i) in inhibiting the enzymatic activity of GST-OTUB2 proteins. **e** OTUB2-IN-1 is able to bind a recombinant OTUB2 protein. Representative response curves of SPR assays are shown. RU, response unit. **f** Inhibitory effects of OTUB2-IN-1 on the cellular level of PD-L1 in LL/2 tumor cells. **g** The binding activity between PD-L1-Flag proteins and GST-OTUB2 proteins in the presence of OTUB2-IN-1 examined by a blocking chemiluminescence immunoassay (CLIA). $n = 3$ biologically independent experiments. **h** Effects of OTUB2-IN-1 (10 μM) on LL/2 tumor cell growth. $n = 3$ biologically independent experiments. **i** Treatment scheme. **j** Treatment with OTUB2-IN-1 reduced LL/2 tumor growth. $n = 7$ mice per group. **k** Treatment with OTUB2-IN-1 reduced tumor growth in C57BL/6 mice implanted with LL/2 tumor cells stably overexpressing PD-L1. $n = 7$ mice per group. **l** Representative images of IHC staining with DAPI, anti-CD8α, anti-GZMB and anti-PD-L1 in vehicle- or OTUB2-IN-1-treated tumors. Scale bars, 20 μm. Quantitative estimates of GZMB⁺ CD8⁺ (**m**) and PD-L1⁺ (**n**) cells in vehicle- or OTUB2-IN-1-treated tumors. Three fluorescent fields for each of the three samples were counted. $n = 9$ biologically independent samples. **o** Representative images of IHC staining with DAPI, anti-YAP, anti-P-Akt and anti-P-p65 in vehicle- or OTUB2-IN-1-treated tumors. Scale bars, 20 μm. **p** Quantitative estimates of YAP⁺, P-Akt⁺ and P-p65⁺ cells in vehicle- or OTUB2-IN-1-treated tumors. $n = 9$ biologically independent samples. The data are presented as the mean ± s.e.m., and $P$ values were calculated using two-way analysis of variance (ANOVA) (**h**, **j**, **k**), except for in (**m**, **n**, **p**), in which $P$ values were calculated by an unpaired two-sided Student's $t$ test. ns, not significant.

Clinical analyses have shown that the expression of OTUB2 is upregulated in various cancer tissues and associated with a poor clinical prognosis in multiple human cancers and have also established the connection between OTUB2 and PD-L1, such that the expression of OTUB2 being positively correlated with that of PD-L1 in LUSC and LUAD. We also demonstrated the connection between OTUB2 and CD8 such that the expression of OTUB2 was inversely correlated with the expression of CD8 in various cancers, especially in LUSC. This motivates the development of OTUB2 as a biomarker for cancer diagnosis and a prognostic factor for multiple human cancers. As such, it is possible that OTUB2 expression could have utility in selecting patients more likely to respond to OTUB2-targeted therapies or tolerate immunotherapies. Overall, the widespread overexpression of OTUB2 in solid tumors, represents a previously unappreciated mechanism whereby tumors evade immune-mediated destruction. Moreover, currently available ICB antibodies aiming to enhance tumor cell recognition by lymphocytes and T cell cytotoxicity, such as anti-PD-1, anti-PD-L1, and anti-CTLA-4, have shown great promise in clinical use[53]. However, low levels of CTLs and low immune activity have been observed in many forms of human cancers, especially colon, breast, prostate and lung cancers, resulting in limited activity of ICB antibodies in these patients[54]. Therefore, we expect that OTUB2-targeted inhibitors can also be translated as single therapeutic agents or developed to potentiate the efficacy of ICB therapy in the near future.

The biological function of OTUB2 related to cancer development seems to be complex. Several studies have shown that OTUB2 can promote cancer progression through multiple mechanisms, such as activation of AKT/mTOR signaling[28], Hippo signaling[29] and NF-κB signaling[30]. Our results provide a new notion that OTUB2 is an immune regulator that targets and inhibits PD-L1 degradation, consequently leading to the promotion of T-lymphocyte-mediated antitumor immunity and suppression of tumor growth. In this study, we found that OTUB2 inhibition may also inhibit cancer progression by suppressing Hippo signaling, AKT/mTOR signaling and NF-κB signaling in addition to the reduction of PD-L1 expression. However, differential regulation of Hippo signaling, AKT/mTOR signaling and NF-κB signaling by OTUB2 inhibition was observed in different tumors. Multiple studies have showed that YAP and/or TAZ activation upregulates PD-L1 at the transcriptional level[55], suggesting that OTUB2 may directly or indirectly regulate PD-L1 through multiple mechanisms. Interestingly, our results showed no inhibitory effect of OTUB2 deficiency on cell growth; however, whether OTUB2 deficiency affects cancer metastasis was not investigated in this study and deserves further investigation. Additionally, although OTUB2 potentially modulates the functions of multiple cellular proteins involved in diverse functions, the other posttranslational regulatory targets of OTUB2 in tumor cells have not yet been elaborated. Therefore, whether OTUB2-mediated immune evasion involves other immune-related genes, was not a focus in this study and should be further investigated. In summary, we demonstrated a general mechanism of immunosuppression in tumor cells mediated through OTUB2-mediated stabilization of PD-L1 (Supplementary Fig. 29). This mode of action is critical for different tumor cells to escape immune surveillance via the PD-L1/PD-1 interaction. Importantly, inhibition of PD-L1 stabilization in tumor cells promotes the tumor-infiltrating CTL response. Thus, targeting OTUB2, as exemplified by OTUB2-IN-1 treatment, may represent a potential strategy for PD-L1-targeted therapy of multiple cancers, especially tumors with high OTUB2 expression.

## Methods

### Ethical statement

The protocol and any procedures involving the care and use of animals in this study were reviewed and approved by the Institutional Animal Care and Use Committee of Xiamen University (approval number: XMULAC20190072) before study initiation. The paraffin-embedded tissues of patients were collected with patient consent and the approval of the Biomedical Ethics Review Committee, Shang Hai Outdo Biotech Company (permit number: SHYJS-CP-1810010 and SHYJS-CP-1610001).

### Cell lines and plasmids

B16-F10, LL/2, CT26.WT, NCI-H358, LoVo, SK-MES-1, NCI-H226, KLN205, MDA-MB-231, NCI-1975, FaDu, BT-549, U-87 MG, Hep-2, HeLa, BT-20, HCC1937 and 293 T cells were purchased from American Type Culture Collection (ATCC, Manassas, VA, USA). MC38 cells were purchased from the National Infrastructure of Cell Line Resources (Beijing, China). B16-F10, LL/2, 293 T, MDA-MB-231 and MC38 cells were cultured in DMEM supplemented with 10% fetal bovine serum (FBS), 1× GlutaMax and 100 μ/ml penicillin–streptomycin. CT26.WT, NCI-H358, NCI-H226, HCC1937, NCI-1975 and BT-549 cells were cultured in RPMI-1640 medium supplemented with 10% FBS, 1× GlutaMax and 100 μ/ml penicillin–streptomycin. SK-MES-1, BT-20, HeLa, Hep-2, FaDu, U-87 MG and KLN205 cells were cultured in MEM medium supplemented with 10% FBS, 1× GlutaMax and 100 μ/ml penicillin–streptomycin. LoVo cells were cultured in F-12K medium supplemented with 10% FBS, 1× GlutaMax and 100 μ/ml penicillin–streptomycin. The ATCC cell lines were authenticated using STR profiling by ATCC. Parental lines and their derivatives were confirmed to be mycoplasma negative using the LookOut® Mycoplasma PCR Detection Kit (Sigma–Aldrich, MP0035). Primary human PBMCs were purchased from Lonza (CC-2702), and primary mouse T cells were purified from the mouse splenocytes by using the EasySep™ Mouse T Cell Isolation Kit (STEMCELL, 19853). All primary cells were grown in RPMI 1640 medium supplemented with 10% FBS, 1× GlutaMax, 100 μ/ml penicillin–streptomycin, 20 mM HEPES, 1 mM sodium pyruvate and 50 μM 2-mercaptoethanol. All mammalian cells were maintained at 37 °C in a humidified incubator with 5% $CO_2$.

Plasmids (pLKO.1, psPAX2, PMD.2 G, pCI-neo-cOVA, lentiCRISPR v2 and lentiCas9-Blast) were obtained from Addgene. Human and mouse OTUB2, mouse PD-L1, and HA-Ub were synthesized by

GENEWIZ and cloned into the lentiCas9-Puro vector containing a puromycin selection cassette. sgRNA-resistant human and mouse OTUB2 (res OTUB2) constructs were synthesized by GENEWIZ and cloned into the lentiCas9-Blast vector containing a blastcidin selection cassette. Flag-OTUB2, OTUB2-Flag and Myc-OTUB2 were cloned into lentiviral vectors. OTUB2 FL and the truncated mutant OTUB2 44-234 (amino acids 44-234) were cloned into lenti-HA vectors, and PD-L1 FL, PD-L1 3NQ mutant (N192Q/N200Q/N219Q) and the truncated mutant PD-L1△ICD (amino acids 1-260) were cloned into lenti-MCS-Flag or lenti-Myc-MCS vectors. The C51S mutation in OTUB2, the K48R, K63R, K48 and K63 mutations in the lenti-HA-Ub construct and the 3KR mutations (K262R/K271R/K280R) in the lenti-Myc-PD-L1 FL construct were introduced by site-directed mutagenesis. GST-tagged OTUB2 or OTUB2 C51S was cloned into the pGEX-4T vector. The strawberry sequence and tumor antigen ovalbumin (OVA) sequence were subcloned into the lentiCas9-Blast vector via Gibson assembly to generate Strawberry and cOVA-expressing vectors (lenti-Blast-cOVA-T2A-Strawberry).

## Mice

C57BL/6 J and NOD-SCID mice (female, 5 or 6 weeks old) were purchased from GemPharmatech (Jiangsu, China), DBA/2 mice (female, 5 or 6 weeks old) were purchased from SLAC Laboratory Animal Co., Ltd. (Shanghai, China), C57BL/6-Tg(TcraTcrb)1100Mjb/J mice (OT-1 mice, strain#: 003831, 5 or 6 weeks old) were purchased from the Jackson Laboratory, and maintained at the specific-pathogen-free animal core facility of the University of Xiamen University. All mice were housed at a temperature of 25 °C in a humidity-controlled environment with free access to food and water in a 12 h light/dark cycle. For animal studies, mice were earmarked before grouping and randomly separated into groups. We used 5 to 12 mice per experimental group in all animal experiments. For MC38, LL/2 or KLN205 cells, $5 \times 10^5$ cells in 100 μl of PBS were injected. For B16-F10 cells, $3 \times 10^5$ cells were injected. To deplete CD8$^+$ T cells, 200 μg of InVivoMAb anti-mouse CD8α (2.43, Bio X Cell, BE0061) or InVivoMAb rat IgG2b (LTF-2, Bio X Cell, BE0090) was intraperitoneally injected 2 days before tumor cell injection, and this treatment was continued every four days until the end of the experiment. To evaluate the efficacy of inhibitors, OTUB2-IN-1 was intraperitoneally injected 6 days after tumor cell injection, and this treatment was continued daily for five days. Tumor volume was measured every 2–3 days using calipers and calculated using the formula Volume = (L × (W)²)/2, where L and W represent the largest and the smallest diameters, respectively. Mice were euthanized via $CO_2$ asphyxiation when the tumors reached 1000 mm³ in volume or became necrotic.

## CRISPR/Cas9-mediated gene knockout

Stable OTUB2 KO in the mouse cell lines MC38, B16-F10, LL/2 and CT26 and human cell lines NCI-H358 and 293 T was generated using lenti-CRISPR v2 vectors containing specific sgRNA produced in 293 T cells and selection with puromycin (1 μg/ml for B16F10, MC38, LL/2 and 293 T cells; 2 μg/ml for CT26 cells) for 5−7 days. The cells were then seeded in 96-well plates. Single clones were then expanded, and the expression of the target protein was detected by western blotting. Finally, editing in the positive clones was validated by performing DNA sequencing of the genomic sequences in the vicinity of the corresponding sgRNA region. An sgRNA sequence targeting a nonspecific target sequence (5′-GAACGACTAGTTAGGCGTGTA-3′) was used as a negative control. The sgRNA sequences targeting mouse OTUB2 were as follows: 1, 5′-GCAAAAGATTCACCTCGATC-3′; and 2, 5′-GGTA-GAGTACGTCGACGAGA-3′. The sgRNA sequence targeting human OTUB2 was as follows: 5′-TTCAACGACCAGAGTGCCT-3′.

## Generation of stable cell lines

Stable KD of OTUB2 in mouse or human tumor cells was generated using lentiviruses produced in 293 T cells with packaging plasmids

(pMD2.G and psPAX2) and pLKO.1 vectors containing specific shRNA against OTUB2. Then, infected cells were selected with puromycin (1 μg/ml for MC38, LoVo, NCI-H358, SK-MES-1, MDA-MB-231 and NCI-H226) for 5 days and the expression of the target protein was detected by western blotting. An shRNA sequence targeting a nonspecific target sequence (5′-CCTAAGGTTAAGTCGCCCTCG-3′) was used as a negative control. The shRNA sequences targeting mouse OTUB2 were as follows: 1, 5′-GATGAGGAGATGGACATCAAA-3′; 2, 5′-CTTCGGTTTATCT GCTCTATA-3′; 3, 5′-GAAGACCAAAGGAGACGGAAA-3′; and 4, 5′-CTAT CCATTCTTCGGGAT CAT-3′. The shRNA sequences targeting human OTUB2 were as follows: 1, 5′-CCTATGTGTCACTGGATTATT-3′; 2, 5′-TG TGGTGGAACTGGTAGAGAA-3′; 3, 5′-CGAGATGGATACCGCCCTGAA-3′; 4, 5′-CATCCCACTACAACATCCTTT-3′; and 5, 5′-TGTGGTGGAACTGG-TAGAGAA-3′. The shRNA sequences targeting other DUBs were provided in Supplementary Table 3.

Stable OE mouse and human tumor cell lines were generated using lentiviruses produced in 293 T cells with packaging plasmids (pMD2G and pspAX2) and lenti-Puro-EV, lenti-Puro-hOTUB2, lenti-Puro-mOTUB2, lenti-Puro-mPD-L1, lenti-Blast-EV, lenti-Blast-res OTUB2, lenti-Blast-res OTUB2 C51S, lenti-Blast-cOVA-T2A-Strawberry or lenti-Blast-EV-Strawberry. Then, infected cells were selected with puromycin for 5 days or blasticidin (6 μg/ml for B16-F10, 2 μg/ml for LL/2 cells and 15 μg/ml for NCI-H358 cells) for 8 days, and the expression of the target protein was detected by western blot.

## Cell viability assay

In total, $2 \times 10^3$ of each of the indicated cell lines were seeded in 96-well plates in triplicate in 100 μl of complete medium per well. After the indicated time points, 10 μl of Cell Counting Kit-8 (CCK-8) cell proliferation assay reagent (MedChemExpress (MCE), HY-K0301) was added to the cells, and the cells were incubated at 37 °C for another 4 h. The optical absorbance was determined at 450 nm using a microplate reader (Thermo Fisher Scientific, Multiskan Go 1510).

## Western blot analysis and co-IP

Cells were washed with cold phosphate-buffered saline (PBS) and then lysed with radioimmunoprecipitation assay (RIPA) buffer supplemented with protease inhibitors (Roche). Equal amounts of lysates mixed with loading buffer were separated by SDS-polyacrylamide gel electrophoresis (PAGE) (SurePAGE™, Bis-Tris, 10 × 8, 4–20%; M00656) and transferred to nitrocellulose membranes. The proteins were then probed with specific primary antibodies followed by horseradish peroxidase (HRP)-conjugated secondary antibodies (Bio-Rad, 1706516, 1:3000; Bio-Rad, 1706515, 1:3000). The primary antibodies were anti-OTUB2 (Sangon Biotech, D199590, 1:500), anti-PD-L1 (Abcam, ab213480, 1:500), anti-PD-L1 (ProteinTech, 66248-1-Ig, 1:500), anti-HA (Sigma–Aldrich, H3663, 1:2000), anti-FLAG (Sigma–Aldrich, F1804, 1:2000), anti-c-Myc (Sigma–Aldrich, C3956, 1:500), anti-β-Actin (ProteinTech, 66009-1-Ig, 1:2,000), anti-GST-HRP (ProteinTech, HRP-66001, 1:500) and anti-GAPDH (ProteinTech, 60004-1-Ig, 1:2000). For immunoprecipitation, cells were lysed with Pierce IP Lysis Buffer (Thermo Fisher Scientific, 87787). The cell lysates were incubated with the indicated antibodies at 4 °C overnight, followed by incubation with Dynabeads® Protein A (Thermo Fisher Scientific, 100-02D) at 4 °C for another 1 hour. Flag- or HA-tagged proteins were precipitated with anti-Flag® M2 Affinity Gel (Sigma–Aldrich, A2220) or Pierce anti-HA Magnetic Beads (Thermo Fisher Scientific, 88836). The coprecipitates were separated using SDS–PAGE and immunoblotted with specific antibodies. The HRP signal was developed with Lumi-Light$^{PLUS}$ Western Blotting Substrate (Roche, 12015196001). Western blot images were captured by an Image Quant LAS4000 chemiluminescence imaging system (GE Healthcare) or a Fusion FX SPECTRA (VILBER). Quantification of proteins of interested was performed using Image J (NIH, version 1.53t). For protein identification by mass spectrometry, cell lysates of B16-F10 cells overexpressing Flag-tagged OTUB2 were

collected and immunoprecipitated with anti-Flag® M2 Affinity Gel (Sigma–Aldrich, A2220). The Flag peptide eluents were analyzed by the mass spectrometry facility of Xiamen University.

## Generation of recombinant GST-OTUB2 protein

In brief, a single plasmid-transformed *E. coli* BL21 (DE3) colony was inoculated into 20 ml of LB medium supplemented with 100 µg/ml ampicillin and grown overnight at 37 °C, 200 r.p.m. The overnight culture was transferred to 1 L of LB medium with ampicillin and grown at 37 °C, 200 r.p.m., for ~3–4 h until $OD_{600}$ reached 0.6. The culture was added with 1 mM IPTG and further incubated at 18 °C, 200 r.p.m., for 12 h. *E. coli* cells were harvested by centrifugation and resuspended in 30 ml PBS (containing 5 mM DTT). Cell suspensions were homogenized with a SONICS Vibra–Cell VCX-800 ultrasonic processor at 4 °C for 12 min and then centrifuged at 25,000 g at 4 °C for 30 min. The GST-OTUB2 protein in clarified lysate was purified using a glutathione sepharose 4B column (GE Healthcare, 17075601). GST protein elution was conducted with 5 mM reduced GSH in 50 mM Tris–HCl buffer (pH 8.0). Purified protein was concentrated and dialyzed in PBS overnight at 4 °C. Protein concentration was measured using a BCA Protein Assay Kit (Pierce, 23225), diluted to 1 mg/ml and stored at −20 °C.

## In vitro and in vivo deubiquitination assays

To analyze the deubiquitination of PD-L1 in vitro, 293 T cells were cotransfected with HA-Ub and Flag-tagged PD-L1. The cells were treated with 20 µM MG132 for 8 h to accumulate ubiquitinated proteins before harvest. Cell lysates were then subjected to immunoprecipitation to obtain ubiquitinated PD-L1, which was enriched using anti-Flag® M2 Affinity Gel (Sigma–Aldrich, A2220). The precipitated proteins were washed three times using deubiquitinating buffer (60 mM HEPES, 5 mM $MgCl_2$, and 4% glycerol; pH 7.6) and stored at −20 °C. Protein concentration was determined using a BCA Protein Assay Kit (Pierce, 23225). 1 µg of purified Flag-tagged ubiquitinated PD-L1 proteins were incubated with 1 µg of recombinant GST-OTUB2 WT proteins or recombinant GST-OTUB2 C51S proteins in deubiquitinating buffer at 37 °C for 4 h. The mixtures were subjected to SDS–PAGE and blotted with anti-ubiquitin (CST, 3936, 1:1000). To evaluate inhibitor potency, 3 µg of purified Flag-tagged ubiquitinated PD-L1 proteins were incubated with 0.3 µg of recombinant GST-OTUB2 WT proteins in deubiquitinating buffer with increasing concentrations of OTUB2-IN-1 at 37 °C for 4 h. The mixtures were subjected to dot blot analysis using Bio-Dot Apparatus (Bio-Rad, 1706545) with the indicated antibodies.

To analyze the deubiquitination of PD-L1 in vivo, vector control and OTUB2-KO 293 T cells were individually cotransfected with HA-Ub and PD-L1-Flag or OTUB2 expression vectors. Seventy-two hours after transfection, the cells were treated with MG132 for 8 h before harvest, and then lysed with RIPA buffer supplemented with protease inhibitors (MCE, HY-K0011) and 1% SDS. After mild sonication, the mixtures were boiled at 95 °C for 10 min. The denatured cell lysates were subjected to immunoprecipitation with anti-Flag® M2 Affinity Gel (Sigma–Aldrich, A2220) or Pierce anti-HA Magnetic Beads (Thermo Fisher Scientific, 88836), followed by western blot analysis with the indicated antibodies.

## In vitro pull-down assay

1 µg of purified Flag-tagged ubiquitinated PD-L1 proteins were incubated with 1 µg of recombinant GST-OTUB2 proteins in PBS with increasing concentrations of OTUB2-IN-1 at 4 °C for 2 h. Next, glutathione sepharose 4B (GE Healthcare, 17075601) or anti-Flag® M2 Affinity Gel (Sigma–Aldrich, A2220) was added to the buffer and the mixture was incubated at 4 °C for 2 h. The precipitated proteins were subjected to western blot analysis with the indicated antibodies.

## Chemiluminescence immunoassay (CLIA)

The effects of OTUB2-IN-1 on blocking the interactions between OTUB2 and PD-L1 was determined by an indirect CLIA. Briefly, 96-well plates were coated with 200 ng/well purified Flag-tagged PD-L1 proteins, and nonspecific binding was blocked with PBS containing 20% calf bovine serum. 1 µg of recombinant GST-OTUB2 proteins were incubated with increasing concentrations of OTUB2-IN-1 at 37 °C for 30 min. The mixtures were added to the coated wells for a 1-h incubation, followed by washing and reaction with an anti-GST antibody (Sino Biological, 11213-RP01) and horseradish peroxidase (HRP)-conjugated secondary antibodies (Bio-Rad, 1706515). After the addition of luminol substrates (Wantai BioPharm) for 5 min, the plates were measured with a chemiluminescence reader (Berthod).

## Ubiquitin-Rhodamine 110 (R110) cleavage assays

Cleavage assays with ubiquitin- R110, a C-terminal derivatization of ubiquitin with R110, were performed essentially as previously described[44]. In brief, 550 ng of recombinant GST-OTUB2 and 0.5 µM Rhodamine 110-Ub (R&D Systems, U-550-050) was incubated with varying concentrations of OTUB2-IN-1 (100, 50 or 25 µM) in 100 µl of HEPES buffer (50 mM HEPES, pH 7.8, 0.5 mM EDTA) at 25 °C for 30 min in 96-well plates (PerkinElmer, 6005558). Deubiquitinating activity was determined by measuring fluorescence every 60 s at 485 nm excitation/535 nm emission using a Tecan Infinite E plex. Relative OTUB2 activity was calculated as a ratio of test sample's fluorescence to the fluorescence of the control sample.

## TUBE (Tandem Ubiquitin Binding Entity) pull-down assay

Ubiquitination of endogenous proteins was determined using a TUBE pull-down assay. Briefly, cells were treated with MG132 (10 µM) for 12 h before lysis. Cells were lysed in TUBE lysis buffer (50 mM Tris, pH 7.5, 150 mM NaCl, 1 mM EDTA, 1% NP-40, 10% glycerol) supplemented with protease inhibitors (Roche), 1,10-phenanthroline (MCE, HY-W004544) (5 mM), N-Ethylmaleimide (Topscience, T3088) (5 mM) and PR-619 (MCE, HY-13814) (100 µM). Cell lysates were clarified by centrifugation and incubated with TUBE2 magnetic beads (Life-Sensors, UM-0402M-1000) at 4 °C for 2 h. To enrich Flag-tagged TUBEs, clarified cell lysates were immunoprecipitated with anti-Flag® M2 Affinity Gel (Sigma–Aldrich, A2220). Beads or gels were then washed three times and boiled with 4× SDS loading buffer. The precipitated proteins were subjected to western blot analysis with the indicated antibodies.

## Duolink in situ proximity ligation assay (PLA)

To detect the interaction between OTUB2 and PD-L1, we used a Duolink® in situ PLA kit (Sigma–Aldrich, DUO92101). Cells were fixed in 4% paraformaldehyde for 10 min at room temperature and subsequently blocked using 1× blocking solution. The cells were then incubated with primary antibodies targeting OTUB2 and PD-L1 at 4 °C overnight, followed by incubation with PLA probes at 37 °C for 1 h. After washing three times, a ligation solution containing the ligase was added and incubated at 37 °C for 30 min. Next, the slides were incubated with an amplification solution containing the polymerase at 37 °C in the dark for 100 min. After final washes, the cells were stained with mounting medium containing 4,6-diamidine-2-phenylindole (DAPI, Invitrogen, D1306). Fluorescence images were captured by a white-light laser confocal microscope (Leica TCS SP8 X) using a 100× objective lens.

## Flow cytometry

For analysis of cell-surface PD-L1 expression, mouse or human tumor cells were treated with or without IFN-γ (NCI-H358, SK-MES-1 and NCI-H226, 0.5 ng/ml; B16F10, CT26, LL/2 and LoVo, 1 ng/ml) for 48 h to stimulate PD-L1 expression and stained with APC-labeled anti-mouse PD-L1 antibodies (BioLegend, 124312) or APC-labeled anti-human PD-L1 antibodies (BioLegend, 329708), respectively, for 20 min on ice. After

washing with PBS, the surface expression of PD-L1 was analyzed using a BD LSRFortessa™ X-20 flow cytometer.

For analysis of the number and function of tumor-infiltrating lymphocytes (TILs), vector control or OTUB2-KO tumors were harvested at 12 days post implantation. Tumor tissues were then digested and processed as described previously[56] and subjected to flow cytometry analysis using fluorescein-conjugated antibodies against CD45 (BioLegend, 103140), CD8α (BioLegend, 100706), IFN-γ (BioLegend, 505810) and granzyme B (BioLegend, 372214). Dead cells were labeled using the Zombie Aqua™ Fixable Viability Kit (BioLegend, 423102). Intracellular markers were stained with the Fixation/Permeabilization Solution Kit (BD, 554714) according to the manufacturer's instructions. Immunostained cells were subsequently subjected to flow cytometry analysis by using a BD LSRFortessa™ X-20 flow cytometer. Cell profiles were first recorded using FACSDIVA software (version 9.0), and the data were then analyzed using FlowJo (version 10.0.7r2).

### CHX chase and protein degradation assays
To determine the effect of OTUB2 on proteasomal degradation of the PD-L1 protein, vector control or OTUB2-OE cells were treated with 50 µg/ml CHX (MCE, HY-12320) to inhibit protein biosynthesis for 0 h, 4 h, 8 h, 12 h, 16 h or 24 h. For proteasome inhibition, vector control and OTUB2-KD or OTUB2-KO tumor cells were treated with 20 µM MG132 (MCE, HY-13259) for 12 h. For lysosome inhibition, vector control and OTUB2-KD or OTUB2-KO tumor cells were treated with 20 µM chloroquine (MCE, HY-17589) for 12 h. For ER inhibition, vector control and OTUB2-KD or OTUB2-KO tumor cells were treated with 20 µM EerI (MCE, HY-110078) for 0 h, 8 h, 12 h or 16 h. The cells were then harvested and PD-L1 proteins were detected by western blot analysis.

### T-cell-mediated killing assay
Activated OT-1 T cells were generated by incubating $2 \times 10^6$ per ml OT-1 T cells in RPMI medium containing 1 µg/ml SIINFEKL peptide (Sangon) and 50 µ/ml mouse recombinant IL-2 (Sino Biological, 51061-MNAE-20). On day 7, the cells were harvested for assays. Vector control and OTUB2-KO tumor cells (expressing the Strawberry fluorescent protein) were first stimulated with 1 ng/ml IFN-γ for 12 h, seeded in 96-well plates overnight and then incubated with or without activated OT-1 T cells for 24 h. The effector-to-target (E:T) ratio between activated cells and tumor cells was set at 1:1. Subsequently, Strawberry fluorescence was determined by flow cytometry. Target cell survival is shown as the normalized cell index (normalized to data collected in the absence of activated OT-1 T cells).

To obtain activated human PBMCs, PBMCs were activated with Dynabeads™ Human T-Activator CD3/CD28 (Thermo Fisher Scientific, 11131D) and cultured in RPMI medium containing 50 U/ml human recombinant IL-2 (STEMCELL, 78036). Tumor cells were first stimulated with 1 ng/ml IFN-γ for 12 h, seeded in 24-well plates overnight and then incubated with or without activated PBMCs for 120 h. The E:T ratio between activated cells and tumor cells was set at 5:1. Cell viability of remaining tumor cells was quantitatively analyzed by flow cytometry.

### Bioinformatic analysis
We investigated the gene transcriptome data of diverse cancer types using TIMER. The Gene_Corr (correlation) module in TIMER2.0 (http://timer.cistrome.org)[57] was used to examine the relationships between OTUB2 differential expression and CD8[+] T-cell-associated genes in all TCGA tumors by using Spearman's correlation test. The Gene-DE (differential gene expression) module in TIMER2.0 was used to test the differential mRNA expression of OTUB2 between tumors and adjacent normal tissues in all cancers. Survival analysis was used to examine the prognostic value of overall survival and disease-free survival associated with OTUB2 differential expression in diverse malignancies in Gene Expression Profiling Interactive Analysis (GEPIA, http:// http://gepia.cancer-pku.cn/index.html)[58]. Processed data for LUSC or LUAD with OTUB2 OE and control expression were downloaded from the UCSC Xena database. Heatmaps were visualized with Phantasus v1.5.1 (https://genome.ifmo.ru/phantasus)[59].

### Tissue microarray
A human LUSC microarray (HLugS180Su01, 90 cancer tissues from patients, and 90 adjacent paracancerous tissues with IHC staining data for CD8 and PD-L1) or a human LUAD microarray (HLugA180Su03, 92 cancer tissues from patients, and 88 adjacent paracancerous tissues with IHC staining data for PD-L1) was purchased from Shang Hai Outdo Biotech Company.

### IHC and IF
To stain cells in culture, cells were seeded in CellCarrier™ 96-well microplates (PerkinElmer, 6005558) at 1000 cells per well and cultured for 24 h. To perform immunostaining, cells were washed with PBS, fixed in 4% paraformaldehyde for 20 min at room temperature, and then blocked in 10% normal goat serum in PBST buffer for 30 min at room temperature. Subsequently, the cells were incubated with the indicated primary antibodies, ER-Tracker Red (Beyotime, E34250), or LysoTracker Red (Beyotime, C1046) at 4 °C overnight. Anti-OTUB2 (Affinity, AF9147), anti-Calnexin (CST, 2679 S) and anti-PD-L1 (ProteinTech, 66248-1-Ig) antibodies were used. The cells were washed again with PBS, incubated with secondary antibodies (Alexa Fluor 488-conjugated donkey anti-mouse IgG, Invitrogen, A-21202; TRITC-conjugated goat anti-rabbit IgG, Sigma–Aldrich, T6778; Alexa Fluor 488-conjugated donkey anti-rabbit IgG, Invitrogen, A-21206; Alexa Fluor 647-conjugated goat anti-mouse IgG, Invitrogen, A-21237) for 1 h in the dark at room temperature and washed with PBS again. Nuclei were counterstained with DAPI. Images of immunostained samples were obtained using a confocal fluorescence microscope (Leica TCS SP8 X) with a 100× objective lens.

To stain formalin-fixed, paraffin-embedded (FFPE) sections, sections (4–6 µm) were deparaffinized, rehydrated and subjected to antigen retrieval. The slides were stained with the indicated antibodies at 4 °C overnight, followed by incubation with secondary antibodies and visualization using DAB as the substrate. Images were obtained using an Aperio ScanScope XT scanner (Leica). All immunostaining images were evaluated blindly by pathologists from Zhongshan Hospital Affiliated to Xiamen University based on the histochemical score. The intensities of OTUB2, PD-L1 and CD8 staining were classified as follows: 0, no staining; 0.5, weak reactivity; 1, moderate reactivity; 2, strong reactivity; and 3, very strong reactivity. IHC results were scored by multiplying the percentage of positive cells (P%) by the intensity (I%) (Formula: Q = P × I, Maximum = 300). Tumors with a staining score less than or equal to 50 were subclassified into the low OTUB2 expression group, and tumors with a staining score greater than 50 were subclassified into the high OTUB2 expression group.

Multiplex staining was performed with the indicated antibodies with subsequent antibody detection using an OPAL 6-PLEX manual detection kit (NEL811001KT, AKOYA), and corresponding fluorophores as well as spectral DAPI were applied following the manufacturer's instructions. Anti-CD8 (Abcam, ab209775), anti-OTUB2 (Affinity, AF9147), anti-GZMB (Abcam, ab4059), anti-PD-L1 (ProteinTech, 17952-1-AP), anti-YAP (Affinity, DF3182), anti-p-AKT1 (Abcam, ab81283) and anti-p-p65 (Affinity, AF2006) antibodies were used. The slides were scanned and captured using a LEICA DM6B. Phenotypic quantifications and analyses were conducted by individuals blinded to the samples and outcomes to assess the percentages of CD8[+] cells, granzyme B[+] cells, PD-L1[+] cells, YAP[+] cells, p-Akt [+] cells and p-p65[+] cells.

### Structure-based virtual screening
Structure-based virtual screening was performed using a hierarchic pipeline as described[60,61]. To perform virtual screening for small-

molecule inhibitors of OTUB2, we obtained the target protein structure (PDB ID: 4FJV) from the Protein Data Bank. The docking region was centered at Cys51 in the catalytic pocket of OTUB2 and was prepared with the Protein Preparation and Grid Preparation Wizard of Schrödinger (Maestro version 11.4) using standard settings. In the first stage, the Life Chemicals HTS Compound Collection containing 494,400 small molecule drugs (LC-HTS) was docked into the catalytic pocket using the HTVS mode of GLIDE. The top 5% scored small molecules were selected based on the best docking scores. In the secondary stage, the selected small molecules were redocked using the same docking program in the standard precision (SP) mode. The top 5% scored small molecules were selected based on the best docking scores. In the tertiary stage, the selected small molecules were redocked using the same docking program in the extra precision mode. The top 20 scored compounds were selected for further evaluation. Each compound was analyzed for its ability to bind to a recombinant OTUB2 protein and deubiquitinate ubiquitinated PD-L1.

### Determination of the binding affinity of OTUB2-IN-1 for OTUB2

The binding affinity of OTUB2-IN-1(Life Chemicals, F0444-0064) for OTUB2 was determined through an SPR assay. Briefly, experiments were performed at 25 °C on a BIAcore 8 K instrument (GE Healthcare) using CM5 sensor chips coated with GST-OTUB2 proteins, and data were analyzed using evaluation software for the BIAcore 8 K instrument following the manufacturer's instructions.

### Statistical analysis

Statistical analyses were performed using GraphPad Prism 7 software (GraphPad Software). Differences were considered significant when the $P$ value was less than 0.05 ($^*P < 0.05$; $^{**}P < 0.01$; $^{***}P < 0.001$; $^{****}P < 0.0001$; ns, not significant). Two-way ANOVA was used for multiple comparisons in tumor growth comparison experiments. Log-rank tests were used for mouse survival analyses. Animal experiments were repeated as indicated in the figure legends. In other experiments, comparisons between two groups were made with unpaired two-sided Student's $t$ tests. Overall survival of the high and low OTUB2 expression groups in the indicated patient cohorts was evaluated using the log-rank test. Spearman's correlation test was used to analyze the relationships between OTUB2 and PD-L1 or CD8A in the indicated patient cohorts.

### Reporting summary

Further information on research design is available in the Nature Portfolio Reporting Summary linked to this article.

## Data availability

Source data are provided with this paper. Unprocessed western blot images are provided as Source Data File. ShRNAs, gRNAs, plasmids, antibodies, reagents and chemicals are listed in the Supplementary Table 3-7. Source data are provided with this paper.

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

## Acknowledgements

The authors are grateful to MedChemExpress LLC, for their technical aid in the virtual screening job described in this work. This work was supported by the National Natural Science Foundation of China (grant no. 81991491 to N.X.).

## Author contributions

N.X., C.H., J.Z. and P.L. designed the studies, interpreted the results and wrote the manuscript; W.R. and Z.X. carried out the biochemical and animal experiments; F.J. and Z.L. prepared all knockout cell clones and knockdown cells and performed statistical analyses; Y.C. and Z.L. carried out all other experiments; H.W. screened the compounds, evaluated the properties of compounds in vitro; M.Z., S.C. and Y.L. conducted part of the animal experiments; N.W. and J.Y. conducted all the IHC experiments; Y.R. assisted with all biochemical experiments; C.X. assisted with screening and evaluation of OTUB2 inhibitors; C.L. analyzed TCGA clinical data.

## Competing interests
