## [Peer Review File · Nature Communications]

Pharmaceutical targeting of OTUB2 sensitizes tumors to cytotoxic T cells via degradation of PD-L1REVIEWER COMMENTS

Reviewer #1 (Remarks to the Author):

Ren et al. characterized OTUB2 as a key regulator of PD-L1 in tumor cells. They showed that OTUB2 interacted with PD-L1 in the ER, OTUB2 decreased the ERAD of PD-L1 to impair T-lymphocyte-mediated antitumor immunity. Moreover, they identified OTUB2-IN-1 as a specific inhibitor of targeting OTUB2 to reduce the protein level of PD-L1 in tumor cells and suppress tumor growth. The studies are basically well designed. However, the molecular mechanism underlying the inhibition of OTUB2-IN-1 on PD-L1 remain to be elucidated, and several points need to be addressed.

Major concerns:

1. Although the authors showed that OTUB2 as a key regulator of PD-L1 in various tumor cells, one or two types of cancer cell lines should be consistently used throughout the study. Especially, because they showed that overexpression of OTUB2 in tumor cells correlated with poor survival in patients with lung squamous cell carcinoma (LUSC), one or two types of LUSC cell lines should be used for different experiments (biochemical, cellular and animal levels), in addition to use other types of cancer cell line to verify their results.

2. Previous studies have already identified several DUBs that regulate PD-L1 stability, it is curious that what are the relationships among these DUBs in modulation of PD-L1? Does different DUBs function specifically in certain types of cancer?

3. A correlated question related to the above one, in Figure 7h and other studies, the protein expression level of PD-L1 is very high in breast cancer cell MDA-MB-231, and in this study, the authors did not show the effect of OTUB2 on PD-L1 in breast cancer cell. Did the authors examine the effect of OTUB2 on PD-L1 in breast cancer cells? Or it had no effect on PD-L1 in breast cancer (the other published studies did not show OTUB2 is a DUB of PD-L1 in breast cancer). A discussion of different DUBs' role in PD-L1 regulation should be discussed.

4. In general, PD-L1 is glycosylated in many cancer cells, as shown in Fig 1d, 1e, 7h, and other publications; why only a sharp single band of PD-L1 was observed in Fig 4I, 6b and other panels?

5. The cell line used in different experiments should be labeled in the figure, in addition to explain in the figure legend, to help the readers understand the detail information.

6. The authors showed that ectopically expressed OTUB2-WT but not catalytically inactive OTUB2 (C51S mutant) associated with PD-L1 in HEK293T cells (Fig. 3e); and ICD (261–290) of PD-L1 in the interaction with OTUB2 and the OTU domain (44–234) of OTUB2 in the interaction with PD-L1 (Fig. 3i, j). These results suggest that Cys51, the key residue of OTUB2 DUB activity is also essential for its interaction to PD-L1. Therefore, OTUB2 affects PD-L1 not only rely on the DUB enzymatic activity but also on its interaction with PD-L1. The authors should discuss this working mechanism, are there other DUBs function similarly?

7. It is interesting to see that OTUB2-IN-1 was able to reduce the PD-L1 level in tumor cells in a dose-dependent manner, but it failed to affect the stability of OTUB2. So what is the molecular mechanism underlying the inhibition of OTUB2-IN-1 on PD-L1? Does OTUB2-IN-1 block the interaction between OTUB2 and PD-L1? The authors should perform experiments to address this question.

Minor

1. Figure 2f, Tumor volume over time in immunodeficient mice implanted with vector control or OTUB2- KO MC38 tumor cells. It is better to label "immunodeficient mice" in the figure in addition to in the legend.

2. In all panels of western blot assay, molecular weight of proteins should be labeled.

3. Figure 3c,d, change "PDL1" to "PD-L1".

4. Figure 4h, "shOTU" should be labeled as "shOTUB2".

Reviewer #2 (Remarks to the Author):

In the submitted work the authors investigate the regulation of PD-L1, a transmembrane protein that suppresses the adaptive immune response, by the DUB OTUB2 in the context of tumor cell survival. The scientific question and the overarching clinical significance of the question being pursued by the authors was logically laid out and concisely described in the introduction.

The authors show that PD-L1 is deubiquitinated by OTUB2 resulting in its stabilization, they identify the domains in each protein responsible for their interaction, and show that PD-L1 is degraded through the ERAD pathway. Taken together, the authors' results highlight the possibility of exploiting high expression of OTUB2 as a biomarker for cancer and/or indicator of tumor aggressiveness, which could inform the type of treatment option(s) chosen. Additionally, the OTUB2 inhibitor OTUB2-IN-1 identified by the authors shows promising potential in targeting OTUB2 as a personalized therapeutic intervention in the treatment of susceptible cancer. The clinical significance of these results warrants further investigation.

While I believe the results are of interest to the scientific community, the manuscript requires considerable revision prior to publication. The length of the manuscript needs to be addressed, specifically the abstract and the fact that there are ten figures, and some figures have 10+ panels. Some of the figures in the main text could be moved to extended data or removed entirely as they do not meaningfully contribute to the story. More thoughtfully designed experiments would allow the authors to convey important findings with fewer panels and thus permit simpler interpretation of the data by the reader. There is inadequate explanation of experimental design related to figures 5 and 6, preventing anyone from successfully reproducing the authors' work (e.g., purification of GST-tagged proteins, protein quantification). There are inconsistencies between main text, figure legends, and labeling of panels in the figures. The authors should explain their reasoning for choosing to investigate only OTUB2 in relation to PD-L1 stability considering there are nearly 100 human DUBs, some of which presumably also act on PD-L1. Finally, I recommend the authors create a supplementary table(s) showing plasmids, shRNA, associated experiments, Addgene identification numbers, and other important reagent information for transparency, accessibility, and reproducibility.

Major issues and questions:

1) The authors did not investigate or mention any other DUBs that may play a role regulating PD-L1 levels. Of the nearly 100 human DUBs, it is reasonable to assume a handful of other DUBs are also involved in regulation. At the very least, the authors should mention why they focused solely on OTUB2 and/or their rationale for excluding other DUBs in their study. Finally, the authors should include additional controls to determine whether Knockdown or Knockout of other DUBs (i.e. OTUB1, USP7, etc) have any effect on PDL1 levels for negative controls.

2) I think it would be worth commenting on similarities/differences of OTUB2 between mice and humans since both models are used throughout without explanation.

3) Figure 5 and 6

-Line 206 describes a ubiquitination assay involving MG132 in Figure 5B, but this is not reflected in the figure panel or in the figure legend.

-The main text states reintroducing OTUB2 into KO cells rescued ability to reduce ubiquitinated PD-L1 (line 207), but this is not obvious in the IP in Figure 5C. By eye, the levels of overall ubiquitination are clearly reduced in OTUB2 KO + OE lanes compared to OTUB2 KO lanes, but the same can't be said for ubiquitinated PD-L1. This reduction is more apparent in Figure 5D.

-The authors state K48 and K63 chains are associated with protein degradation and only look at these two linkages despite modern consensus in the literature that many chain linkages can trigger proteasomal degradation. The MG132 experiments better demonstrate that PD-L1 is degraded by the proteasome. Fig 5E, F do not meaningfully contribute to the results that PD-L1 is stabilized by OTUB2.

-MG132 and chloroquine are introduced in the main text without any explanation of their function or pathways they inhibit

4) The methods section lacks sufficient description of how proteins were purified and quantified (specifically recombinant GST-OTUB2) to allow replication of the in vitro deubiquitination assays described in Figures 5 and 6.

5) Under what circumstances is PD-L1 degraded through the ERAD pathway? Is PD-L1 ubiquitinated and removed from the membrane because it is misfolded? Does OTUB2 de-ubiquitinate native PD-L1, misfolded PD-L1, or both? Does OTUB2 also de-ubiquitinate other transmembrane proteins? This may be beyond the scope of the manuscript, but it would bring helpful context to the overall function of OTUB2 in the cell.

Minor issues and questions:

1) The level of granularity in the abstract makes it unnecessarily lengthy and is better suited to the introduction in the main body text.

2) Line 82, authors states that recent studies have demonstrated tumor cells may upregulate PD-L1 expression following PD-L1 antibody treatment, yet only one study is referenced.

3) Figure 2C, legend icons are shown but not labeled as done in other panels in the figure.

4) Why are there references shown as et al? It would be helpful to see all listed authors.

5) Line 166, when describing the catalytically inactive form of OTUB2, C51S is more useful than simply CS.

6) Figure 3J shows two bands for each OTUB form in the input HA blot. What are the bands? Other forms of OTUB2 that do not bind PD-L1?

7) Figure 4A, D flow data could be moved to extended as the panels are difficult to decipher by eye and thus not particularly useful in the main figure.

8) Figure 4K and M are purportedly quantifying the same kind of assay, yet the y-axis labels convey different measures.

9) Line 209, I would substitute 'often' for 'typically' and provide a reference.

-Additionally, there are only 42 references in the paper, an oddly low amount for the length of the paper.

10) Line 247, there is no figure reference supporting the claim that reduced PD-L1 staining was observed in OTUB2-KO tumor cells.

11) Line 367, states "recent studies" have developed inhibitors for DUBs, but the referenced articles are from 2018 and 2011, not what one would consider recent.

Reviewer #3 (Remarks to the Author):

The manuscript by Ren et al. reports the role of OTUB2, a deubiquitinating enzyme, in regulating PD-L1 in a wide variety of tumor cells. The authors demonstrated that OTUB2 interacts with PD-L1, deubiquitinates, and stabilizes PD-L1 protein by inhibiting its proteosomal degradation independent of IFN γ stimulation. They also identified OTUB2-IN-1, a specific inhibitor of OTUB2, which reduces the expression of PD-L1 in tumor cells and suppresses tumor growth by promoting intratumoral infiltration of cytotoxic T-Lymphocytes. The data presented in this study are clear, well-presented, and adequate controls are included in the experiments. A few specific concerns are raised as mentioned below.

1) The biochemical data showing OTUB2 interaction with PD-L1 and regulation is strong. But the effect of OTUB2 underexpression or overexpression in regulating the surface expression of

endogenous or IFN γ -induced PD-L1 is weak. Is it due to the fact that PD-L1 is regulated by multiple mechanisms? As a result, the effect of the mutant (OTUB2 C51S) in impairing PD-L1 expression is also weak.

2) Most of the ubiquitination experiments with endogenous proteins were performed only in one cell line (mouse melanoma B16-F10 cells). To establish a generalized mechanism for the effect of OTUB2 on PD-L1 stabilization, it is better to use at least two cell lines throughout the study, which the authors are using for example, MC38, H-358, CT26, etc.

3) It is not clear why the authors used only lung squamous cell carcinoma, and not lung adenocarcinoma, to test the association between elevated expression of OTUB2 and overall survival, PD-L1/CD8 protein expression. As lung adenocarcinoma is involved in NSCLC and these patients are benefited by anti-PD1/PD-L1 therapy, these changes by OTUB2 in adenocarcinoma should be addressed.

4) The effect of the inhibitor OTUB2-IN-1 on PD-L1 expression was shown only in Lewis Lung carcinoma LL/2 and B16-F10 cell line. To make a broad statement about the effect of the inhibitor it is better to use multiple cell lines that are being used in this study.

5) The effect of the inhibitor OTUB2-IN-1 on tumor growth and survival in Lewis Lung cancer model is modest. This compound should also be tested in vivo using other cell lines.

6) OTUB2 promotes cancer progression by regulating Hippo, NF-kB and Akt/mTOR pathways. The inhibitor OTUB2-IN-1 may also inhibit tumor growth by suppressing those pathways in addition to the regulation of PD-L1 expression. Moreover, the regulation of YAP/TAZ in the Hippo Pathway by OTUB2 may induce PD-L1, thus leading to CTL regulation (Nguyen and Yi, 2019). Therefore, it will be informative to test the activation of few markers of those pathways in the tumors from OTUB2-IN-1 treated mice using western blotting or immunohistochemistry.

In summary, interesting data are presented regarding the stabilization of PD-L1 by OTUB2 that leads immune-suppression in tumor cells and promotes tumor growth. However, addressing the concerns mentioned above and performing the relevant experiments will strengthen the studies.

REVIEWER COMMENTS

Reviewer #1 (Remarks to the Author):

Ren et al. characterized OTUB2 as a key regulator of PD-L1 in tumor cells. They showed that OTUB2 interacted with PD-L1 in the ER, OTUB2 decreased the ERAD of PD-L1 to impair T-lymphocyte-mediated antitumor immunity. Moreover, they identified OTUB2-IN-1 as a specific inhibitor of targeting OTUB2 to reduce the protein level of PD-L1 in tumor cells and suppress tumor growth. The studies are basically well designed. However, the molecular mechanism underlying the inhibition of OTUB2-IN-1 on PD-L1 remain to be elucidated, and several points need to be addressed.

Response: Thank you for your positive comments and valuable suggestions to improve the quality of our manuscript.

Major concerns:

1. Although the authors showed that OTUB2 as a key regulator of PD-L1 in various tumor cells, one or two types of cancer cell lines should be consistently used throughout the study. Especially, because they showed that overexpression of OTUB2 in tumor cells correlated with poor survival in patients with lung squamous cell carcinoma (LUSC), one or two types of LUSC cell lines should be used for different experiments (biochemical, cellular and animal levels), in addition to use other types of cancer cell line to verify their results.

Response: Thanks for the reviewer's suggestion. In addition to a human lung adenocarcinoma (NCI-H358) and a murine lung cancer cell line (LL/2), we also used two human lung squamous cell carcinoma cell lines (SK-MES-1 and NCI-H226) to evaluate the regulation of PD-L1 by OTUB2.

2. Previous studies have already identified several DUBs that regulate PD-L1 stability, it is curious that what are the relationships among these DUBs in modulation of PD-L1? Does different DUBs function specifically in certain types of cancer?

Response: Thanks for the reviewer's suggestion. At least five DUBs were reported to

stabilize PD-L1 in different cancers, including USP7, USP9X, USP22, CSN5 and OTUB1. USP7, USP9X and USP22 can stabilize PD-L1 in gastric cancer, oral squamous cell carcinoma or liver cancer, respectively^{1, 2, 3}. The regulation of PD-L1 degradation by CSN5 or OTUB1 is mainly found in breast cancer^{4, 5}. We performed additional experiments to investigate the potential of these five previously reported DUBs in regulating PD-L1 levels in two NSCLC cancer cell lines.

We demonstrated OTUB2-mediated PD-L1 regulation as a common feature in various human and mouse cancers, particularly in NSCLC. We found OTUB2 to be the top positive regulator of PD-L1 protein stability in NSCLC, since OTUB2 depletion showed the most significant impact on PD-L1 abundance. However, other five DUBs function differently in regulating PD-L1 abundance in NSCLC cells. Knockdown of USP22 or OTUB1 only significantly reduced PD-L1 abundance in NCI-H358 cells but not in SK-MES-1 cells. Meanwhile, the PD-L1 protein level remained unchanged or slightly changed after knockdown of CSN5, USP9X or USP7 in NSCLC cells. Why OTUB2 functions differently from other DUBs in regulating PD-L1 abundance in NSCLC or other cancers needs to be further studied in the future.

We have added these results and also included a discussion of the roles of these DUBs in PD-L1 regulation in our revised manuscript (**Fig. 3g and Extended Data Fig. 10a-g, Line 199-205, Line 423-435**).

3.A correlated question related to the above one, in Figure 7h and other studies, the protein expression level of PD-L1 is very high in breast cancer cell MDA-MB-231, and in this study, the authors did not show the effect of OTUB2 on PD-L1 in breast cancer cell. Did the authors examine the effect of OTUB2 on PD-L1 in breast cancer cells? Or it had no effect on PD-L1 in breast cancer (the other published studies did not show OTUB2 is a DUB of PD-L1 in breast cancer). A discussion of different DUBs' role in PD-L1 regulation should be discussed.

Response: Thanks for the reviewer's suggestion. We carried out new experiments to test the effect of OTUB2 on PD-L1 in MDA-MB-231 cells. The effects of OTUB2 deficiency on the downregulation of PD-L1 levels were also observed in MDA-MB-231 cells. One recent study did not show the effect of OTUB2 on the regulation of PD-

L1 possibly due to the low knockdown efficiency of OTUB2, in fact, the knockdown efficiency of the shRNA targeting OTUB2 was not shown in that study.

We have added some new data and included a discussion of these DUBs' roles in PD-L1 regulation in the revised manuscript (**Extended Data Fig. 8c, Line 191-192, Line 423-435**).

4. In general, PD-L1 is glycosylated in many cancer cells, as shown in Fig 1d, 1e, 7h, and other publications; why only a sharp single band of PD-L1 was observed in Fig 4I, 6b and other panels?

Response: Thanks for your comments. In fact, PD-L1 is glycosylated in all cancer cells we used, although the degree of glycosylation varies in different cancer cells. In our study, we used precast polyacrylamide gels with a gradient of increasing acrylamide concentration (4-20%) for SDS-PAGE analysis. The acrylamide gradient in the resolving gel reduces diffusional band spreading. Leading molecules are slowed down by the higher concentration, lagging molecules accelerated by the lower. If the target proteins stayed in the gels with the higher concentration, more sharper bands will be observed. For consistency, we have repeated the experiments with adjusted gel running time and obtained the blot image with more visible glycosylated PD-L1. We have replaced the previous results with them (**Fig. 3i, l and Extended Data Fig. 13a, b**).

5. The cell line used in different experiments should be labeled in the figure, in addition to explain in the figure legend, to help the readers understand the detail information.

Response: Thanks for your suggestion, the cell lines used in different experiments were clearly labeled in our revised manuscript.

6. The authors showed that ectopically expressed OTUB2-WT but not catalytically inactive OTUB2 (C51S mutant) associated with PD-L1 in HEK293T cells (Fig. 3e); and ICD (261–290) of PD-L1 in the interaction with OTUB2 and the OTU domain (44–234) of OTUB2 in the interaction with PD-L1 (Fig. 3i, j). These results suggest that Cys51, the key residue of OTUB2 DUB activity is also essential for its interaction to PD-L1. Therefore, OTUB2 affects PD-L1 not only rely on the DUB enzymatic activity but also on its interaction with PD-L1. The authors should discuss this working mechanism, are there other DUBs function similarly?

Response: Thanks for your suggestions. In our study, we found that the C51 residue of OTUB2, which is located in the OTU domain (44–234) of OTUB2 in the interaction with the ICD of PD-L1, not only affects its enzymatic activity but also is involved in its interaction with PD-L1. This is similar with a previous study showing that OTUB2 C51S loses its binding ability to STAT1 and fails to deubiquitinate STAT1⁶. However, several reports have described different modes of action of DUBs in regulating their substrates, such as deubiquitination of YAP/TAZ by OTUB2⁷ and deubiquitination of PD-L1 by CSN5⁵, the key residues responsible for the enzymatic activity of DUBs are not essential for DUBs–substrate interactions. OTUB2 C51S can still interact with YAP/TAZ through the interaction between the SUMO site K233 in the OTU domain and the SUMO interaction motifs of YAP/TAZ, but fails to deubiquitinate YAP/TAZ⁷. Another DUB CSN5 specifically interacts with the PD-L1 and directly deubiquitinates PD-L1⁵. The enzyme active center (MPN domain) of CSN5 is not responsible for its interaction to PD-L1, but disrupting the MPN domain affects CSN5-mediated PD-L1 deubiquitination. We have discussed different modes of action of DUBs in regulating their substrates in our revised manuscript (**Line 406-418**).

7. It is interesting to see that OTUB2-IN-1 was able to reduce the PD-L1 level in tumor cells in a dose-dependent manner, but it failed to affect the stability of OTUB2. So what is the molecular mechanism underlying the inhibition of OTUB2-IN-1 on PD-L1? Does OTUB2-IN-1 block the interaction between OTUB2 and PD-L1? The authors should perform experiments to address this question.

Response: Thanks for your suggestion. We carried out new experiments to ensure that the ability of OTUB2-IN-1 to reduce PD-L1 expression was due to the impaired deubiquitinase activity of OTUB2 rather than the impaired binding of OTUB2 to PD-L1 induced by OTUB2-IN-1. We first tested the ability of OTUB2-IN-1 to inhibit the hydrolytic activity of recombinant GST-OTUB2 protein toward its universal substrate Ub. Incubation of recombinant GST-OTUB2 protein with OTUB2-IN-1 resulted in a significant inhibition of Ub-R110 cleavage by GST-OTUB2 in a dose-dependent manner, suggesting that OTUB2-IN-1 can inhibit the hydrolytic activity of OTUB2 toward its substrate Ub-R110 *in vitro*. We further verified the ability of OTUB2-IN-1

to block the binding of OTUB2 to PD-L1 using a co-IP assay. We found that OTUB2-IN-1 did not interfere with protein interactions between OTUB2 and PD-L1. Our results indicate that OTUB2-IN-1 can regulate PD-L1 protein abundance through inhibition of OTUB2 catalytic activity rather than blocking the interactions between OTUB2 and PD-L1. These experiments also indicate that OTUB2-IN-1 relies on the catalytic activity of OTUB2 to regulate PD-L1 abundance.

We have added these results and also included a discussion in our revised manuscript (**Fig. 9g, Extended Data Fig. 19c, d and Extended Data Fig. 21, Line 359-375, Line 452-454**).

Minor

1. Figure 2f, Tumor volume over time in immunodeficient mice implanted with vector control or OTUB2- KO MC38 tumor cells. It is better to label “immunodeficient mice” in the figure in addition to in the legend.

Response: Thanks for your suggestion. We added the labels in Fig. 1f of the revised manuscript.

2. In all panels of western blot assay, molecular weight of proteins should be labeled.

Response: Thanks for your suggestion, protein molecular weight markers were clearly labeled in our revised manuscript.

3. Figure 3c,d, change “PDL1” to “PD-L1”.

Response: Thanks for your suggestion, this wrong labeling has been revised (**Fig. 2d and Extended Data Fig. 7c-e**).

4. Figure 4h, “shOTU” should be labeled as “shOTUB2”.

Response: Thanks for your suggestion, this wrong labeling has been revised (**Fig. 3h**).

Reviewer #2 (Remarks to the Author):

In the submitted work the authors investigate the regulation of PD-L1, a transmembrane protein that suppresses the adaptive immune response, by the DUB OTUB2 in the context of tumor cell survival. The scientific question and the overarching clinical significance of the question being pursued by the authors was logically laid out and

concisely described in the introduction.

The authors show that PD-L1 is deubiquitinated by OTUB2 resulting in its stabilization, they identify the domains in each protein responsible for their interaction, and show that PD-L1 is degraded through the ERAD pathway. Taken together, the authors' results highlight the possibility of exploiting high expression of OTUB2 as a biomarker for cancer and/or indicator of tumor aggressiveness, which could inform the type of treatment option(s) chosen. Additionally, the OTUB2 inhibitor OTUB2-IN-1 identified by the authors shows promising potential in targeting OTUB2 as a personalized therapeutic intervention in the treatment of susceptible cancer. The clinical significance of these results warrants further investigation.

While I believe the results are of interest to the scientific community, the manuscript requires considerable revision prior to publication. The length of the manuscript needs to be addressed, specifically the abstract and the fact that there are ten figures, and some figures have 10+ panels. Some of the figures in the main text could be moved to extended data or removed entirely as they do not meaningfully contribute to the story. More thoughtfully designed experiments would allow the authors to convey important findings with fewer panels and thus permit simpler interpretation of the data by the reader. There is inadequate explanation of experimental design related to figures 5 and 6, preventing anyone from successfully reproducing the authors' work (e.g., purification of GST-tagged proteins, protein quantification). There are inconsistencies between main text, figure legends, and labeling of panels in the figures. The authors should explain their reasoning for choosing to investigate only OTUB2 in relation to PD-L1 stability considering there are nearly 100 human DUBs, some of which presumably also act on PD-L1. Finally, I recommend the authors create a supplementary table(s) showing plasmids, shRNA, associated experiments, Addgene identification numbers, and other important reagent information for transparency, accessibility, and reproducibility.

Response: Thank you for your positive comments and valuable suggestions to improve

the quality of our manuscript. We have made extensive modifications to our manuscript and make our manuscript concise and clear. Some data in our previous manuscript were deleted or provided in Supplemental data. We carefully corrected the inconsistencies between main text, figure legends, and labeling of panels in the figures. We also prepared supplementary tables to present all important information on plasmids, sequences, reagents, etc. (**Extended Data Table 3-7**). We have responded specifically to some of important suggestions below.

Major issues and questions:

1) The authors did not investigate or mention any other DUBs that may play a role regulating PD-L1 levels. Of the nearly 100 human DUBs, it is reasonable to assume a handful of other DUBs are also involved in regulation. At the very least, the authors should mention why they focused solely on OTUB2 and/or their rationale for excluding other DUBs in their study. Finally, the authors should include additional controls to determine whether Knockdown or Knockout of other DUBs (i.e. OTUB1, USP7, etc) have any effect on PDL1 levels for negative controls.

Response: Thanks for your suggestions. The rationale for focusing on OTUB2 was based on the bioinformatic correlation analyses specifically for the CD8⁺ T-cell signature gene CD8A with 103 DUB genes⁸ using The Cancer Genome Atlas (TCGA) data sets. We added the rationale and related bioinformatic data in the revised manuscript (**Extended Data Fig. 1a, b and Extended Data Table 1, Line 108-117**).

We also performed additional experiments to investigate the potential of five previously reported DUBs in regulating PD-L1 levels in two NSCLC cancer cell lines. At least five DUBs were reported to stabilize PD-L1 in different cancers, including USP7, USP9X, USP22, CSN5 and OTUB1. USP7, USP9X and USP22 can stabilize PD-L1 in gastric cancer, oral squamous cell carcinoma or liver cancer, respectively^{1, 2, 3}. The regulation of PD-L1 degradation by CSN5 or OTUB1 is mainly found in breast cancer^{4, 5}. We performed additional experiments to investigate the potential of these five previously reported DUBs in regulating PD-L1 levels in two NSCLC cancer cell lines. We demonstrated OTUB2-mediated PD-L1 regulation as a common feature in various

human and mouse cancers, particularly in NSCLC. We found OTUB2 to be the top positive regulator of PD-L1 protein stability in NSCLC, since OTUB2 depletion showed the most significant impact on PD-L1 abundance. However, other five DUBs function differently in regulating PD-L1 abundance in NSCLC cells. Knockdown of USP22 or OTUB1 only significantly reduced PD-L1 abundance in NCI-H358 cells but not in SK-MES-1 cells. Meanwhile, the PD-L1 protein level remained unchanged or slightly changed after knockdown of CSN5, USP9X or USP7 in NSCLC cells. Why OTUB2 functions differently from other DUBs in regulating PD-L1 abundance in NSCLC or other cancers needs to be further studied in the future.

We have added these results and also included a discussion of the roles of these DUBs in PD-L1 regulation in our revised manuscript (**Fig. 3g and Extended Data Fig. 10a-g, Line 199-205, Line 423-435**).

2) I think it would be worth commenting on similarities/differences of OTUB2 between mice and humans since both models are used throughout without explanation.

Response: Thanks for your suggestions. Homogeneous analysis demonstrates that amino acid sequences of *OTUB2* gene between mouse and human share 94.9% homology and are highly conservative. Because of their highly conserved catalytic OTU domain, OTUB2 homologs from human and mouse may display conserved function that can specifically remove Ub chains from target proteins⁹. To better examine the immunoregulatory function of OTUB2 in vivo, we used syngeneic mouse models (e.g., MC38, LL/2 and B16F10 tumor cell lines), which provide an effective approach for studying how cancer therapies perform in the presence of a functional immune system. We have added the explanations in the Results section (**Line 135-138**).

3) Figure 5 and 6

-Line 206 describes a ubiquitination assay involving MG132 in Figure 5B, but this is not reflected in the figure panel or in the figure legend.

Response: Thanks for your suggestions. We have added the information in the related figure panels and figure legends (**Fig. 4, Fig. 5 and Extended Data Fig. 12**).

-The main text states reintroducing OTUB2 into KO cells rescued ability to reduce ubiquitinated PD-L1 (line 207), but this is not obvious in the IP in Figure 5C. By eye,

the levels of overall ubiquitination are clearly reduced in OTUB2 KO + OE lanes compared to OTUB2 KO lanes, but the same can't be said for ubiquitinated PD-L1. This reduction is more apparent in Figure 5D.

Response: Thanks for your comments. In Figure 5C, we performed co-IP experiments to obtain total PD-L1-Flag protein, the HA signals represent the ubiquitinated PD-L1-Flag protein. A smear of slower migrating bands was observed for PD-L1-Flag protein, indicative of polyubiquitin chain formation. It should be more accurate to describe the 'ubiquitinated PD-L1' as 'polyubiquitinated PD-L1'. Monoubiquitination has largely been linked to chromatin regulation, protein sorting, and trafficking, whereas polyubiquitination has typically been associated with protein proteasomal degradation^{10, 11}. For better interpretation, we rephrased the statements describing Figure 4a-i, Figure 5e,f in our revised manuscript (**Line 214-259**).

-The authors state K48 and K63 chains are associated with protein degradation and only look at these two linkages despite modern consensus in the literature that many chain linkages can trigger proteasomal degradation. The MG132 experiments better demonstrate that PD-L1 is degraded by the proteasome. Fig 5E, F do not meaningfully contribute to the results that PD-L1 is stabilized by OTUB2.

Response: Thanks for your comments and suggestions. We agreed the MG132 experiments in Figure 5A better demonstrated that PD-L1 is degraded by the proteasome. Due to the formation of PD-L1 polyubiquitination, we are curious about which kind of polyubiquitin chains that are involved in the proteasome-dependent degradation of PD-L1. Previous evidence has shown that OTUB2 cleaved Lys48 and Lys63 chains preferentially^{12, 13} and OTUB2 stabilized its target KRT80 through both Lys-48-linked and Lys-63-linked deubiquitination¹⁴, so we investigated whether polyubiquitin chains with a Lys-48 linkage or a Lys-63 linkage were involved in the proteasome-dependent degradation of PD-L1. Our results in Figure 5E and 5F demonstrated that OTUB2 deubiquitinates PD-L1 through Lys-48-linked but not Lys-63-linked deubiquitination, which should help us to understand how OTUB2 regulates PD-L1 deubiquitination. We have added these information in our revised manuscript (**Line 227-231**).

-MG132 and chloroquine are introduced in the main text without any explanation of their function or pathways they inhibit

Response: Thanks for your suggestions. We have added the function descriptions about MG132 and chloroquine in the revised manuscript (**Line 217 and Line 242**).

4) The methods section lacks sufficient description of how proteins were purified and quantified (specifically recombinant GST-OTUB2) to allow replication of the in vitro deubiquitination assays described in Figures 5 and 6.

Response: Thanks for your suggestions. We have added the detailed descriptions about recombinant GST-OTUB2 and other related proteins in the Methods section (**Line 636-649, Line 658-666 and Line 676-702**).

5) Under what circumstances is PD-L1 degraded through the ERAD pathway? Is PD-L1 ubiquitinated and removed from the membrane because it is misfolded? Does OTUB2 de-ubiquitinate native PD-L1, misfolded PD-L1, or both? Does OTUB2 also de-ubiquitinate other transmembrane proteins? This may be beyond the scope of the manuscript, but it would bring helpful context to the overall function of OTUB2 in the cell.

Response: Thanks for your suggestions. In eukaryotes, cellular homeostasis requires the ubiquitin-dependent degradation of membrane proteins, which are mainly degraded either by endoplasmic reticulum-associated degradation (ERAD) or by endosomal sorting complexes required for transport (ESCRT)-dependent lysosomal degradation¹⁵. The ERAD pathway not only uses to clear misfolded proteins from the ER for cytosolic proteasomal degradation but also maintains appropriate levels of membrane-associated proteins¹⁶.

N-linked glycosylation profoundly affects protein folding, dysregulated N-linked glycosylation can lead to misfolded ER proteins and premature degradation¹⁷. We asked whether OTUB2 can induce the deubiquitination of non-glycosylated (premature or misfolded) PD-L1 protein. We found that OTUB2 suppressed polyubiquitination of both non-glycosylated and glycosylated PD-L1. In this study, we demonstrate that OTUB2 is likely to regulate the physiological quantity of PD-L1 proteins via reducing

ubiquitination of both premature and mature PD-L1. We have added these results and also included a discussion in our revised manuscript (**Fig. 4e, Line 224-227, Line 414-418**).

We also carried out additional experiments to investigate the relations between OTUB2 and other transmembrane proteins, such as the transmembrane protein CD300A. We found that CD300A can only be monoubiquitinated in HEK293 cells, which is different from the polyubiquitination of PD-L1 protein. OTUB2 cannot deubiquitinate CD300A possibly due to CD300A is not the substrate of OTUB2.

Minor issues and questions:

1) The level of granularity in the abstract makes it unnecessarily lengthy and is better suited to the introduction in the main body text.

Response: Thanks for your suggestion, the abstract have been revised (**Line 31-47**).

2) Line 82, authors states that recent studies have demonstrated tumor cells may upregulate PD-L1 expression following PD-L1 antibody treatment, yet only one study is referenced.

Response: Thanks for your comments. We have added one more reference in the revised manuscript (**Line 72**).

3) Figure 2C, legend icons are shown but not labeled as done in other panels in the figure.

Response: Thanks for your suggestion, the legends have been clearly labeled in the revised manuscript (**Extended Data Fig. 4c and Fig. 9k**).

4) Why are there references shown as et al? It would be helpful to see all listed authors.

Response: Thanks for your suggestion, Nature Communications uses standard Nature referencing style. All authors should be included in reference lists unless there are six or more, in which case only the first author should be given, followed by 'et al.'.

5) Line 166, when describing the catalytically inactive form of OTUB2, C51S is more useful than simply CS.

Response: Thanks for your suggestion, we have substituted 'CS' with 'C51S' in the revised manuscript.

6) Figure 3J shows two bands for each OTUB form in the input HA blot. What are the bands? Other forms of OTUB2 that do not bind PD-L1?

Response: Thanks for your comments. Regarding the additional bands in the input HA blot, we thought they are highly likely the non-specific bands recongnized by anti-HA antibodies or partial degradation of the HA-tagged protein. When the whole lysates were purified by Flag pulldown assay and subjected to western blot, the non-specific bands or degraded bands can be significantly reduced (Left panel in Fig. 2g). We repeated this experiment, similar results were observed.

7) Figure 4A, D flow data could be moved to extended as the panels are difficult to decipher by eye and thus not particularly useful in the main figure.

Response: Thanks for your suggestion. 'Flow data' were made as Supplemental data or Source data in our revised manuscript.

8) Figure 4K and M are purportedly quantifying the same kind of assay, yet the y-axes labels convey different measures.

Response: Thank you for pointing this out. The y-axes labels in previous Figure 4k and 4m were different but expressed the same meaning. To be consistent, The y-axes labels have been revised (**Fig. 3k and 3m**).

9) Line 209, I would substitute ‘often’ for ‘typically’ and provide a reference.

-Additionally, there are only 42 references in the paper, an oddly low amount for the length of the paper.

Response: Thanks for the reviewer’s suggestion. We have substituted ‘often’ with ‘typically’ and provided a reference for this statement (**Line 228**). We have checked the literature carefully and added more references to better support our manuscript (61 references in total).

10) Line 247, there is no figure reference supporting the claim that reduced PD-L1 staining was observed in OTUB2-KO tumor cells.

Response: Thanks for the reviewer’s suggestion. We have added the figure reference for this claim in the revised manuscript (**Line 269**).

11) Line 367, states “recent studies” have developed inhibitors for DUBs, but the referenced articles are from 2018 and 2011, not what one would consider recent.

Response: Thanks for the reviewer’s suggestion. We have revised this sentence and also supplemented some recent references in the revised manuscript (**Line 446-447**).

Reviewer #3 (Remarks to the Author):

The manuscript by Ren et al. reports the role of OTUB2, a deubiquitinating enzyme, in regulating PD-L1 in a wide variety of tumor cells. The authors demonstrated that OTUB2 interacts with PD-L1, deubiquitinates, and stabilizes PD-L1 protein by inhibiting its proteosomal degradation independent of IFN γ stimulation. They also identified OTUB2-IN-1, a specific inhibitor of OTUB2, which reduces the expression of PD-L1 in tumor cells and suppresses tumor growth by promoting intratumoral infiltration of cytotoxic T-Lymphocytes. The data presented in this study are clear, well-presented, and adequate controls are included in the experiments. A few specific concerns are raised as mentioned below.

Response: Thank you for your positive comments and valuable suggestions to improve the quality of our manuscript.

1) The biochemical data showing OTUB2 interaction with PD-L1 and regulation is strong. But the effect of OTUB2 underexpression or overexpression in regulating the surface expression of endogenous or IFN γ -induced PD-L1 is weak. Is it due to the fact that PD-L1 is regulated by multiple mechanisms? As a result, the effect of the mutant (OTUB2 C51S) in impairing PD-L1 expression is also weak.

Response: Thanks for the reviewer's comments. We agreed that PD-L1 in tumor cells is regulated by multiple mechanisms. At least five DUBs were reported to stabilize PD-L1 across different cancers, including USP7, USP9X, USP22, CSN5 and OTUB1. Although USP7, USP22 and OTUB1 may play a certain role in the regulation of PD-L1 in NSCLC, we found that OTUB2 has the most significant impact on the regulation of PD-L1 in NSCLC. We performed additional experiments to investigate the potential of these five previously reported DUBs in regulating PD-L1 levels in two NSCLC cancer cell lines. We have added these results and also included a discussion of the roles of these DUBs in PD-L1 regulation in our revised manuscript (**Fig. 3g and Extended Data Fig. 10a-g, Line 199-205, Line 423-435**).

OTUB2 C51S not only lost its enzymatic ability but also lost its binding ability to PD-L1, so overexpression of OTUB2 C51S failed to impair PD-L1 expression. We have discussed this mode of action of OTUB2 in regulating PD-L1 in our revised manuscript (**Fig. 2c, Line 167-169, Line 406-414**).

2) Most of the ubiquitination experiments with endogenous proteins were performed only in one cell line (mouse melanoma B16-F10 cells). To establish a generalized mechanism for the effect of OTUB2 on PD-L1 stabilization, it is better to use at least two cell lines throughout the study, which the authors are using for example, MC38, H-358, CT26, etc.

Response: Thanks for the reviewer's suggestion. In addition to mouse B16-F10 cell line, we also used a human LUAD cell line NCI-H358, a human LUSC cell line SK-MES-1 and a mouse lung cancer cell line LL/2 to evaluate the effects of OTUB2 on PD-L1 stabilization.

3) It is not clear why the authors used only lung squamous cell carcinoma, and not lung adenocarcinoma, to test the association between elevated expression of OTUB2 and

overall survival, PD-L1/CD8 protein expression. As lung adenocarcinoma is involved in NSCLC and these patients are benefited by anti-PD1/PD-L1 therapy, these changes by OTUB2 in adenocarcinoma should be addressed.

Response: Thanks for the reviewer's suggestion. We found *OTUB2* showed negative associations with the CD8⁺ T-cell signature gene *CD8A* and *OTUB2* were significantly upregulated in the cancer tissues compared with adjacent paracancerous tissues in the TCGA-LUSC cohort, thus we focused on studying the functions of *OTUB2* in LUSC. For further strengthen our study, we also assessed the correlations between *OTUB2* and PD-L1 protein expression in human LUAD specimens. Similar results were also observed in LUAD patients. We also found that LUAD patients with high levels of *OTUB2* exhibited significantly poorer overall survival and *OTUB2* protein levels were positively correlated with PD-L1 expression in human LUAD specimens. We have added these results in our revised manuscript (**Extended Data Fig. 18, Line 317-333**).

4) The effect of the inhibitor OTUB2-IN-1 on PD-L1 expression was shown only in Lewis Lung carcinoma LL/2 and B16-F10 cell line. To make a broad statement about the effect of the inhibitor it is better to use multiple cell lines that are being used in this study.

Response: Thanks for the reviewer's suggestion. We carried out new experiments to test the effect of OTUB2-IN-1 in additional cell lines, including three lung cancer cell lines and one murine LUSC cell line. We found that OTUB2-IN-1 can significantly reduce PD-L1 expression in multiple lung cancer cells. We have added these results in our revised manuscript (**Extended Data Fig. 20b, 20c and Line 375-377**).

5) The effect of the inhibitor OTUB2-IN-1 on tumor growth and survival in Lewis Lung cancer model is modest. This compound should also be tested in vivo using other cell lines.

Response: Thanks for the reviewer's suggestion. We used a murine melanoma cell line B16F10 and a murine LUSC cell line KLN205 to evaluate the anti-tumor effects of OTUB2-IN-1. We found that OTUB2-IN-1 also exhibited strong growth-suppressive effects on LUSC and melanoma. We have added these results in our revised manuscript (**Extended Data Fig. 24, 25**).

6) OTUB2 promotes cancer progression by regulating Hippo, NF- κ B and Akt/mTOR pathways. The inhibitor OTUB2-IN-1 may also inhibit tumor growth by suppressing those pathways in addition to the regulation of PD-L1 expression. Moreover, the regulation of YAP/TAZ in the Hippo Pathway by OTUB2 may induce PD-L1, thus leading to CTL regulation (Nguyen and Yi, 2019). Therefore, it will be informative to test the activation of few markers of those pathways in the tumors from OTUB2-IN-1 treated mice using western blotting or immunohistochemistry.

Response: Thanks for the reviewer's suggestion. We have performed immunohistochemistry in the tumors from OTUB2-IN-1 treated mice to dissect the effects of OTUB2-IN-1 on regulating Hippo, NF- κ B and Akt/mTOR pathways. In LL/2 tumors, in addition to significant inhibition of PD-L1 expression, OTUB2-IN-1 treatment significantly reduced the expression of YAP and phosphorylated p65, but had no effect on the expression of phosphorylated Akt. In B16-F10 tumors, in addition to significant inhibition of PD-L1 expression, OTUB2-IN-1 treatment significantly reduced the expression of phosphorylated Akt, but had no effect on the expression of YAP and phosphorylated p65. In KLN205 tumors, in addition to significant inhibition of PD-L1 expression, OTUB2-IN-1 treatment significantly reduced the expression of phosphorylated p65, but had no effect on the expression of YAP and phosphorylated Akt. These results indicated that OTUB2 may possess more general roles in regulation of PD-L1 in various tumor cell lines. Multiple studies have showed that YAP and/or TAZ activation upregulates PD-L1 at the transcriptional level¹⁸, suggesting that OTUB2 may directly or indirectly regulate PD-L1 through multiple mechanisms. We have added these results and related discussions in our revised manuscript (**Fig. 9o, p; Extended Data Fig. 26, 27; Line 385-394 and Line 480-490**).

In summary, interesting data are presented regarding the stabilization of PD-L1 by OTUB2 that leads immune-suppression in tumor cells and promotes tumor growth. However, addressing the concerns mentioned above and performing the relevant experiments will strengthen the studies.

Response: We appreciate your constructive comments and suggestions. We have made every effort to address these concerns.

Reference

1. Jingjing W, Wenzheng G, Donghua W, Guangyu H, Aiping Z, Wenjuan W. Deubiquitination and stabilization of programmed cell death ligand 1 by ubiquitin-specific peptidase 9, X-linked in oral squamous cell carcinoma. *Cancer Med* **7**, 4004-4011 (2018).
2. Wang Z, *et al.* Abrogation of USP7 is an alternative strategy to downregulate PD-L1 and sensitize gastric cancer cells to T cells killing. *Acta Pharm Sin B* **11**, 694-707 (2021).
3. Huang X, *et al.* USP22 Deubiquitinates CD274 to Suppress Anticancer Immunity. *Cancer Immunol Res* **7**, 1580-1590 (2019).
4. Zhu D, *et al.* Deubiquitinating enzyme OTUB1 promotes cancer cell immunosuppression via preventing ER-associated degradation of immune checkpoint protein PD-L1. *Cell Death Differ* **28**, 1773-1789 (2021).
5. Lim SO, *et al.* Deubiquitination and Stabilization of PD-L1 by CSN5. *Cancer Cell* **30**, 925-939 (2016).
6. Chang W, *et al.* OTUB2 exerts tumor-suppressive roles via STAT1-mediated CALML3 activation and increased phosphatidylserine synthesis. *Cell Rep* **41**, 111561 (2022).
7. Zhang Z, *et al.* OTUB2 Promotes Cancer Metastasis via Hippo-Independent Activation of YAP and TAZ. *Mol Cell* **73**, 7-21 e27 (2019).
8. Harrigan JA, Jacq X, Martin NM, Jackson SP. Deubiquitylating enzymes and drug discovery: emerging opportunities. *Nat Rev Drug Discov* **17**, 57-78 (2018).
9. Du J, Fu L, Sui Y, Zhang L. The function and regulation of OTU deubiquitinases. *Front Med* **14**, 542-563 (2020).
10. Husnjak K, Dikic I. Ubiquitin-binding proteins: decoders of ubiquitin-mediated cellular functions. *Annu Rev Biochem* **81**, 291-322 (2012).
11. Finley D. Recognition and processing of ubiquitin-protein conjugates by the proteasome. *Annu Rev Biochem* **78**, 477-513 (2009).
12. Mevissen TE, *et al.* OTU deubiquitinases reveal mechanisms of linkage specificity and enable ubiquitin chain restriction analysis. *Cell* **154**, 169-184 (2013).
13. Altun M, *et al.* The human otubain2-ubiquitin structure provides insights into

the cleavage specificity of poly-ubiquitin-linkages. *PLoS One* **10**, e0115344 (2015).

14. Ouyang S, *et al.* OTUB2 regulates KRT80 stability via deubiquitination and promotes tumour proliferation in gastric cancer. *Cell Death Discov* **8**, 45 (2022).
15. Schmidt O, *et al.* Endosome and Golgi-associated degradation (EGAD) of membrane proteins regulates sphingolipid metabolism. *Embo J* **38**, (2019).
16. Printsev I, Curiel D, Carraway KL. Membrane Protein Quantity Control at the Endoplasmic Reticulum. *J Membrane Biol* **250**, 379-392 (2017).
17. Wang M, Kaufman RJ. Protein misfolding in the endoplasmic reticulum as a conduit to human disease. *Nature* **529**, 326-335 (2016).
18. Nguyen CDK, Yi C. YAP/TAZ Signaling and Resistance to Cancer Therapy. *Trends Cancer* **5**, 283-296 (2019).

REVIEWER COMMENTS

Reviewer #1 (Remarks to the Author):

The authors addressed most of my concerns. One more question, because OTUB2 promotes cancer progression by regulating Hippo, NF- κ B and Akt/mTOR pathways as well, the inhibitor OTUB2-IN-1 may also inhibit tumor growth by suppressing these pathways in addition to its regulation on PD-L1 expression, therefore, ideally, a rescue experiment should be conducted to verify that OTUB2-IN-1 negatively regulates tumor growth by reducing the abundance of PD-L1. Besides, in Figure 5a and 5b, the OTUB2 in lane 1 and Lane 4 is not knocked down, it seemed that shOTUB2 is mislabeled.

Reviewer #2 (Remarks to the Author):

The authors have now adequately address most, if not all, of the reviewers concerns and comments. The revised manuscript is significantly improved from its original version, and now I find suitable for publication.

Reviewer #3 (Remarks to the Author):

This manuscript has been revised as per suggestions of this reviewer. New data has been added in response to the reviewer's comments that potentially improved the manuscript. Other reviewers' comments were also answered thoroughly through either discussion or experiments. As a result, this manuscript should be accepted for publication in Nature Communications considering other requirements are satisfied.

REVIEWER COMMENTS

Reviewer #1 (Remarks to the Author):

The authors addressed most of my concerns. One more question, because OTUB2 promotes cancer progression by regulating Hippo, NF-kB and Akt/mTOR pathways as well, the inhibitor OTUB2-IN-1 may also inhibit tumor growth by suppressing these pathways in addition to its regulation on PD-L1 expression, therefore, ideally, a rescue experiment should be conducted to verify that OTUB2-IN-1 negatively regulates tumor growth by reducing the abundance of PD-L1. Besides, in Figure 5a and 5b, the OTUB2 in lane 1 and Lane 4 is not knocked down, it seemed that shOTUB2 is mislabeled.

Response: Thank you for your positive comments and valuable suggestions to improve the quality of our manuscript.

- 1) We performed additional experiments to verify that OTUB2-IN-1 negatively regulates tumor growth by reducing the abundance of PD-L1. To clarify the effect of OTUB2-IN-1 in tumor inhibition through regulation of PD-L1 *in vivo*, LL/2 cells with stably expressed PD-L1 and control plasmids, as confirmed in **Extended Data Fig. 24a**, were used to establish tumor models. As expected, in comparison to the control group, overexpressed PD-L1 resulted in a substantial increase of tumor growth, whereas OTUB2-IN-1 significantly reduced the tumor growth. We also found that the antitumor effect of OTUB2-IN-1 could be affected by PD-L1 overexpression (Fig. 9k and Extended Data Fig. 24, b-d). This rescue experiment indicates that OTUB2-IN-1 attenuates tumor growth through regulation of PD-L1. We have added these results in our revised manuscript (**Fig. 9k and Extended Data Fig. 24, b-d, Line 379-388**).
- 2) The legends in Fig. 5a and 5b have been clearly labeled in the revised manuscript (**Fig. 5**).

Reviewer #2 (Remarks to the Author):

The authors have now adequately address most, if not all, of the reviewers concerns

and comments. The revised manuscript is significantly improved from its original version, and now I find suitable for publication.

Response: Thank you for your positive comments and valuable suggestions to improve the quality of our manuscript.

Reviewer #3 (Remarks to the Author):

This manuscript has been revised as per suggestions of this reviewer. New data has been added in response to the reviewer's comments that potentially improved the manuscript. Other reviewers' comments were also answered thoroughly through either discussion or experiments. As a result, this manuscript should be accepted for publication in Nature Communications considering other requirements are satisfied.

Response: Thank you for your positive comments and valuable suggestions to improve the quality of our manuscript.

REVIEWERS' COMMENTS

Reviewer #1 (Remarks to the Author):

The authors addressed all my concerns. The revised manuscript is significantly improved and I think it is suitable for publication.

To editor,

Thanks very much for your kind work and consideration on publication of our manuscript.

REVIEWERS' COMMENTS

Reviewer #1 (Remarks to the Author):

The authors addressed all my concerns. The revised manuscript is significantly improved and I think it is suitable for publication.

Response: Thank you for your positive comments and valuable suggestions to improve the quality of our manuscript.